

# Trends in air pollutants and health impacts in three Swedish cities over the past three decades

Henrik Olstrup[1], Bertil Forsberg[3], Hans Orru[3,4], Mårten Spanne[5], Hung Nguyen[6], Peter Molnár[7], Christer Johansson[1,2]

[1]Atmospheric Science Unit, Department of Environmental Science and Analytical Chemistry, Stockholm University, 11418 Stockholm, Sweden
[2]Environment and Health Administration, SLB, Box 8136, 104 20 Stockholm, Sweden
[3]Division of Occupational and Environmental Medicine, Department of Public Health and Clinical Medicine, Umeå University, 90187 Umeå, Sweden
[4]Department of Family Medicine and Public Health, University of Tartu, 500 90 Tartu, Estonia
[5]Environment department, City of Malmö, 205 80 Malmö
[6]Environmental Administration in Gothenburg, Box 7012, 402 31, Gothenburg, Sweden
[7]Occupational and Environmental Medicine, Sahlgrenska University Hospital & University of Gothenburg, Medicinaregatan 16A, SE-40530 Gothenburg, Sweden

*Correspondence to*: Henrik Olstrup (henrik.olstrup@aces.su.se)

**Abstract.** Air pollution concentrations have been decreasing in many cities in the developed countries. We have estimated time trends and health effects associated with exposure to $NO_x$, $NO_2$, $O_3$, and $PM_{10}$ in the Swedish cities of Stockholm, Gothenburg, and Malmo from the 1990's to 2015. Trend analyses of concentrations have been performed by using the Mann-Kendall test and the Theil-Sen method. Measured concentrations are from central monitoring stations representing urban background levels, and they are assumed to indicate changes in long-term exposure to the population. However, corrections for population exposure have been performed for $NO_x$, $O_3$, and $PM_{10}$ in Stockholm, and for $NO_x$ in Gothenburg. For $NO_x$ and $PM_{10}$, the concentrations at the central monitoring stations are shown to overestimate exposure when compared to dispersion model calculations of spatially resolved population-weighted exposure concentrations, while the reverse applies to $O_3$. The trends are very different for the pollutants that are studied; $NO_x$ and $NO_2$ have been decreasing in all cities, $O_3$ exhibits an increasing trend in all cities, and for $PM_{10}$, there is a slowly decreasing trend in Stockholm, a slowly increasing trend in Gothenburg, and no significant trend in Malmo. When the trends are divided into weekdays and weekends, the decreasing trends associated with $NO_x$ and $NO_2$ are more prominent during weekdays compared to weekends, indicating that local emission reductions from traffic to a large part have contributed to these declining trends.

Health effects in terms of changes in life expectancy are calculated based on the trends in exposure to $NO_x$, $NO_2$, $O_3$, and $PM_{10}$, and the relative risks associated with exposure to these pollutants. The decreased levels of $NO_x$ are estimated to increase the life expectancy by up to 11 months for Stockholm and 12 months for Gothenburg. This corresponds to up to one fifth of the total increase in life expectancy (54–70 months) in the cities during the period 1990–2015. In contrast to $NO_x$ and $NO_2$, the changing trends associated with $O_3$ and $PM_{10}$ have relatively little impact on the change in life expectancy. $NO_x$ and $NO_2$ are



highly associated with vehicle exhaust emissions, indicating that decreasing road-traffic emissions have had significant impact on the public health in these cities.

## 1. Introduction

Air pollution exposure is clearly recognised as an important global risk factor for the development of a large number of diseases and disabilities (Cohen et al., 2017). Considering the ranking of risk factors on a global basis for the total deaths from all causes for all ages and sexes in 2015, exposure to $PM_{2.5}$ (particles with an aerodynamic diameter smaller than 2.5 micrometer) ending in a fifth place worldwide among all risk factors for premature deaths, where smoking, diet, and high blood pressure are included. Exposure to $PM_{2.5}$ in 2015 is expected to have caused 4.2 million premature deaths among the world's population. This can be compared to smoking, with a corresponding value of 6.4 million premature deaths for the year 2015 (State of Global Air, 2017).

In view of the major health impact that exposure to air pollutants appears to be, it is of great importance to analyse the trends in air pollution concentrations, and how these affect the public health.

Worldwide, many cities, especially in high-income countries, show substantial decreasing trends in air pollution concentrations (e.g. WHO, 2016a; Geddes et al., 2016; Colette et al., 2011). Considering the changes in the total amount of emissions in Europe and in the U.S., respectively, there are different conditions. In Europe (EU–27), during the period 2002–2011, the emissions of nitrogen oxides ($NO_x$ = sum of NO and $NO_2$) have decreased by 27 %, and the emissions of primary $PM_{10}$ (particles with an aerodynamic diameter smaller than 10 micrometer) have decreased by 14 % (Guerreiro et al., 2014). In the U.S., the $NO_x$ emissions in eight cities have decreased between 25 % and 48 % during the period 2005–2012 (Tong et al., 2015). Considering the trends in the U.S. regarding all sector-specific emissions during the period 1990–2010, $NO_x$ exhibits a decrease of 48 %, while $PM_{10}$ exhibits a decrease of 50 % (Xing et al. 2013). So, the $NO_x$ and $PM_{10}$ emissions have decreased almost equally in the U.S., while in Europe, the emissions regarding $NO_x$ exhibit a sharper decline compared to $PM_{10}$ (Guerreiro et al., 2014).

For ozone ($O_3$) in Europe, the concentrations tend to increase at urban sites, especially during the cold season, as observed by e.g. Sicard et al. (2016) for cities in France, and for most cities in Europe by Colette et al. (2011). This rise can be explained by increases in imported $O_3$ by long-range transport, and also by a decreased titration by nitrogen monoxide (NO), due to the reduction in local $NO_x$ emissions (Colette et al., 2011; Sicard et al., 2016). Furthermore, considering different statistical metrics, the trends for $O_3$ at both urban and rural sites in the UK are different during the period 1993–2011, depending on the metric that is used; mean and median trends are positive, while the maximum trend is negative (Munir et al., 2013). In Northern Alberta in Canada, where measurement results from four urban locations were analysed, the mean concentrations of $O_3$ have increased at most stations during the period 1998–2014 (Bari and Kindziersky, 2016).

This means that the general population exposure, and presumably also associated health effects of the urban populations, have been reduced for $NO_x$ and $PM_{10}$, but increased for the mean $O_3$ concentrations. So far, there are, however, rather few studies that have assessed the net health gain or health loss associated with the trends. Henschel et al. (2012) has examined



intervention studies focusing on improvements in air quality, and associated health benefits for the assessed population. Some studies have focused on trends and effects associated with particulate matter (Tang et al., 2014; Keuken et al., 2011; Correria et al., 2013), and some studies have also included ozone (Fann and Risley, 2013; Gramsch et al., 2006). Correria et al. (2013) estimated that for the most urban communities in the USA, as much as 18 % of the increase in life expectancy from 2000 to 2007 was attributable to the reduction in $PM_{2.5}$. When health improvements associated with $PM_{2.5}$ trends are compared to those for $O_3$, the health benefits associated with $PM_{2.5}$ are much greater (Fann and Risley, 2013).

In a few studies, air quality dispersion models together with gridded population data and exposure-response functions have been used to assess health impacts of changing air pollutant emissions. E.g. for Rotterdam (The Netherlands), the health benefits associated with decreasing trends for EC (elemental carbon) and $PM_{10}$ have been calculated for the period 1985–2008, and the average gain in life expectancy was 13 and 12 months per person for $PM_{10}$ and EC, respectively (Keuken et al., 2011). Fann and Risley, (2013) estimated changes in ozone and $PM_{2.5}$ concentrations in the U.S. during the period 2000–2007, and they estimated the impact on the number of premature deaths by using health-impact functions based on short-term relative risk estimates for $O_3$ and long-term relative risk estimates for $PM_{2.5}$. Data from monitoring stations were spatially interpolated. Overall, they found net benefits in the number of premature deaths ranging from 22 000 to 60 000 for $PM_{2.5}$, and from 880 to 4 100 for ozone, but interestingly, they found opposing trends in premature mortality associated with $PM_{2.5}$ and ozone at some locations, and a considerable year to year variation in the number of premature deaths. Tang et al. (2014) evaluated the health benefits associated with coal-burning factory shutdowns and accompanied decreasing levels of $PM_{10}$ in Taiyuan in China during the period 2001–2010. They used $PM_{10}$ measurements from monitoring stations, but without considering spatial variations in the exposures, and they used the Chinese national standard of 40 µg m$^{-3}$ as a threshold level. The number of premature deaths associated with $PM_{10}$ levels dropped from 4 948 in 2001 to 2 138 in 2010, and they did not include other pollutants in their analysis.

The studies that have been referred above have used different methodologies of estimating the exposure; by simply taking the concentration at the monitoring stations, and making spatial interpolations between several monitoring stations, or by using dispersion modelling to estimate the spatial distribution of the exposure concentrations. They have also used different relative risks for mortality, and different ways to apply the baseline mortality and the health-effect exposure threshold.

The objective of this study is to quantify and compare the changes in life expectancy, resulting from changes in different pollutants, namely $NO_x$, $NO_2$, $O_3$, and $PM_{10}$, based on measurements during 25 years in the three largest cities in Sweden. We show that the health impacts related to change in different pollutants are different, and we aim to discuss how different trends in different air pollutants affect the life expectancy assessment.



## 2. Methods

### 2.1 The choice of air pollutants for trend analysis

Our main trend analyses are based on simultaneous, continuous measurements of $NO_x$, $NO_2$, $O_3$, and $PM_{10}$, during the period 1990 to 2015 in Stockholm, Gothenburg, and Malmo; the three largest cities in Sweden. Based on these trends, we calculate

the health impacts associated with changes in the population exposure. The changes in life expectancy are calculated with the mean values and the 95 % confidence intervals of the relative risks, while for the trends and the population-weighted exposure concentrations, only the median and the mean values, respectively, have been used, but without considering their confidence intervals. This means that the ranges of the confidence intervals, presented in Table 2, are narrower than they would have been if the confidence intervals of the trend-lines and the population-weighted exposure concentrations were also included in the

calculations. $NO_2$, $O_3$, and $PM_{10}$ have been regulated in the EU directives, and these regulations also set methods, quality assurance, and control (EU, 2008), making these pollutants relevant to analyse in our study.

### 2.2 Measuring sites and instrumentation

In all analysed cities, the measuring site is located in the city center and represents urban background (Fig. 1). Stockholm is the capital and the largest city in Sweden. It is located on the east coast at 59° N latitude and 18° E longitude. Stockholm has a

temperate climate with four distinct seasons. Gothenburg is the second largest city located on the west coast of Sweden at 57° N latitude and 12° E longitude. It is like Stockholm located within the west-wind belt, and the proximity to the Atlantic Ocean means a slightly milder climate compared to Stockholm. Malmo is located in the southernmost part of Sweden at 55° N latitude and 13° E longitude. General information about the three cities regarding population structure, life expectancy at birth, and baseline mortality is presented in Table A1.

The measuring station in Stockholm is located at Torkel Knutssonsgatan on a roof 20 m above ground level, in Gothenburg it is located on a roof 30 m above ground level in the neighbourhood Östra Nordstan, and in Malmo it is located on the roof of the city hall (Rådhuset) in the city center (Fig. 1). In all stations, $PM_{10}$ has been measured by using Tapered Element Oscillating Microbalance (TEOM 1400A, Thermo Fisher Scientific, USA). However, in Gothenburg, the Continuous Dichotomous Ambient Air Monitor (Thermo Scientific 1405-DF TEOM) has also been used. Nitrogen oxides are measured in all stations

by using chemiluminescense (AC 32M, Environnement SA., France). In Stockholm, ozone is measured by using UV-absorption (O342M, Environnement S.A, France). In Gothenburg, ozone has been measured by using non-dispersive UV photometry (EC9811, Ecotech Pty Ltd, Australia), and also by using CLD 700 AL, Ecophysics, Switzerland, and in recent years by using T200, Teledyne, USA. In Malmo, ozone has been measured by using UV-absorption (Thermo Environmental Instruments Model 49C, USA).



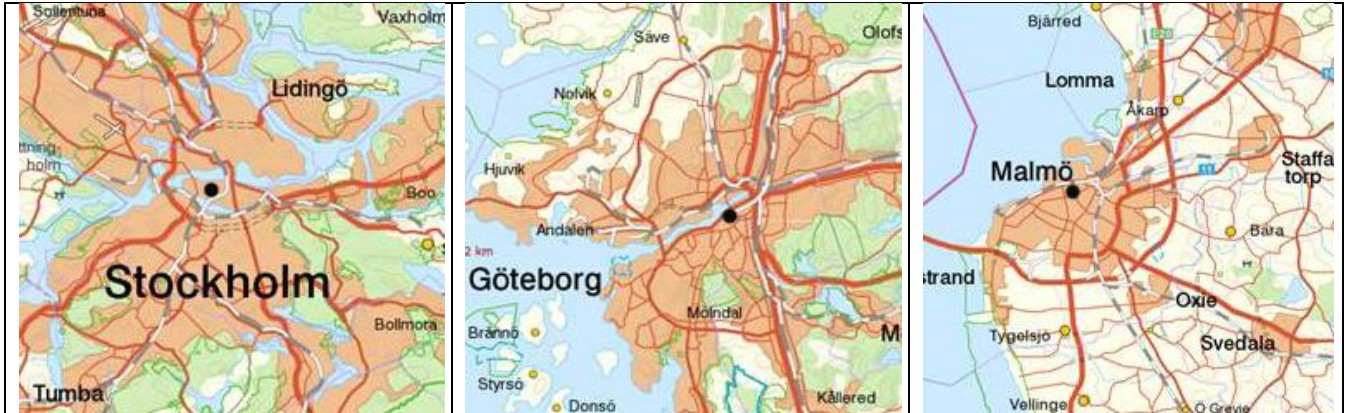

**Figure 1:** Stockholm, Gothenburg, and Malmo; the three largest cities in Sweden, and their surroundings. The black dot in each city shows the location of the measuring station. Each map represents an area of 35 x 35 km.

## 2.3 Statistical analysis of the trends

The changes in air pollution concentrations during the period 1990–2015 have been calculated by using the Openair package (Carslaw and Ropkins, 2012). For the trend analyses, the Mann-Kendall test and the Theil-Sen method have been used. The Mann Kendall test is a non-parametric trend test, which is based on the ranking of observations (Hirsch et al. 1982). The Theil-Sen method is used to calculate the median slope of all possible slopes that may occur between the data points (Theil, 1992; Sen, 1968). In our calculations, the trends are based on monthly averages, and they are adjusted for seasonal variations, as these can have a significant effect on monthly data. The Theil-Sen method is regarded as more suitable than the linear-regression method, as it gives more accurate confidence intervals with non-normal distributed data, and it is not affected as much by outliers.

## 2.4 Health impact and life expectancy calculations

Trends in urban background concentrations reflect the change in population exposure over time (see further below), and they are used to calculate changes in health impacts, presented as changes in life expectancy. As basis for the calculations, we use the population size, age distribution, and mortality rate in Stockholm, Gothenburg, and Malmo according to the year 1997, where data in different age groups have been taken from Statistics Sweden (SCB, 2017), and from the Swedish National Board of Health and Welfare (Socialstyrelsen, 2017). Mortality data before 1997 are not available. However, our own test runs have shown that the calculations in life expectancy give very similar results regardless of the year (1997–2015) in which the population structure and mortality statistics are based on. To illustrate the health benefits associated with decreasing trends, the increase in life expectancy at birth has been calculated in AirQ+ according to a log-linear function (Eq. 1). Similarly, the decrease in life expectancy at birth associated with increasing trends has been calculated in AirQ+ according to the same log-linear function (Eq. 1) (WHO, 2016b). We have applied relative risks obtained from previous epidemiological studies, where





the relationships between mortality and exposure to $NO_x$, $NO_2$, $O_3$, and $PM_{10}$ have been analysed. The concept relative risk (RR) represents a log-linear exposure-response function, where the ratio of the incidence in an exposed group is compared to the incidence in a non-exposed, or a less exposed, group. The mortality rate associated with a change in exposure to pollutant x is calculated as:

$$RR(x) = e^{\beta(X-X0)} \tag{1}$$

where the beta-coefficient ($\beta$) indicates the linear relationship between the health impact and the change ($x - x_0$) in exposure.

For $NO_x$, we apply the RR 1.06 (95 % CI 1.03–1.09) per 10 µg m$^{-3}$ increase, based on the results from Stockfelt et al. (2015), representing all-cause mortality associated with long-term exposure to $NO_x$ in a cohort with men in Gothenburg.

For $NO_2$, we apply the RR 1.066 (95 % CI 1.029–1.104) per 10 µg m$^{-3}$ increase. This RR is based on pooled estimates of mortality associated with long-term exposure to $NO_2$ (Faustini et al., 2014).

For $O_3$, we apply the RR 1.02 (95 % CI 1.01–1.04) per 10 ppb increase, corresponding to 1.01 (95 % CI 1.005–1.02) per 10 µg m$^{-3}$ at 25 ºC and 1 atm, based on a large prospective study examining the associations between long-term ozone exposure, and all-cause and cause-specific mortality (Turner et al., 2016).

For $PM_{10}$, we apply the RR 1.04 (95 % CI 1.00–1.09) per 10 µg m$^{-3}$ increase, which is based on a meta-analysis of 22 European cohorts (Beelen et al., 2014). This value is also in line with the RR estimate of 1.043 (95 % CI 1.026–1.061) per 10 µg m$^{-3}$ increase, since many years used in impact assessments (Künzli et al., 2000), based on cohort studies in the U.S.

All RRs described above are based on calculations for the population aged 30 years and over, and therefore, the changes in life expectancy, calculated in AirQ+, are also based on the age group 30 years old and over. The calculations have also been performed by assuming no threshold under which no effect occurs.

## 2.5 Relationship between urban-background concentrations and population-weighted exposure concentrations

The trends measured at urban background monitoring stations may not be representative for the trends in exposure to the entire population in those areas, due to the position of the urban-background measuring stations, the spatial variation in air pollution concentrations, and the population density within the cities. In order to assess the health effects of the population, associated with changes in air pollution exposure in each metropolitan area, we need to estimate the relations between the concentrations at the urban-background monitoring stations, and the population-weighted exposure concentrations. This is assessed by comparing model-calculated annual population-weighted exposure concentrations with annual-mean urban-background concentrations. We have done this for $NO_x$ in Stockholm and Gothenburg, and for $O_3$ and $PM_{10}$ in Stockholm, but due to lack of data, these relations have not been possible to calculate other than for the above-mentioned pollutants and cities.

Geographically resolved annual-mean concentrations of $NO_x$ in Stockholm were calculated by using a wind model and a Gaussian air-quality dispersion model as a part of the Airviro system (Airviro, 2017). Details on the modelling and emission data are described in Johansson et al. (2017). The same modelling of $NO_x$ was done for Gothenburg by using a Gaussian model (Aermod, US EPA) as a part of the EnviMan AQ Planner (OPSIS, Furulund, Sweden) as described in Molnár et al. (2015).




Spatially resolved $O_3$ concentrations in Stockholm are calculated from a combination of measurements and dispersion modelling of $NO_x$ concentrations. The modelled $NO_x$ concentrations are converted to $O_3$ based on the measured $NO_x$ concentrations at an urban background site in Stockholm, and the difference between $O_3$ at an urban background site and at a rural background site. The basic chemistry behind this is that the $O_3$ concentration within a city is controlled by the transport

from the surrounding areas into the city, and by the removal of $O_3$ due to the reaction with NO (nitrogen monoxide). It is further set that if $O_3$ at the urban background is higher compared to the rural background, then it is set to the same value as the urban background, and if it becomes less than zero, it is set to zero. More details of this method is described in Olsson et al. (2016).

Population-weighted exposure concentrations ($C_{pop}$) are obtained by multiplying the calculated concentration ($C_i$) in each

grid-square cell with the number of people in the corresponding grid-square cell ($P_i$), and summing all products, and dividing the sum by the total population (Eq. 2). This procedure has been used in several previous studies (e.g. Johansson et al., 2009; Orru et al., 2015).

$$Cpop = \frac{\sum Ci\, Pi}{\sum Pi} \qquad (2)$$

The relationships between urban-background levels and population-weighted exposure concentrations are presented in

Appendix A Fig. A1–A4.

## 3. Results

### 3.1 Overview of trends

Fig. 2–4 show the trends in concentrations of $NO_x$, $NO_2$, $O_3$, and $PM_{10}$ measured at urban background sites in Stockholm, Gothenburg, and Malmo during the period 1990–2015. During the given time periods, $NO_x$ and $NO_2$ exhibit decreasing trends

in all cities, whereas $O_3$ exhibit increasing trends in all cities, and for $PM_{10}$, the trends are less clear and consistent. For $PM_{10}$ in Stockholm and Malmo, the data from the measuring stations only include the period 1997–2015 and 1996–2015, respectively. In several cases, the trends are not perfectly linear throughout the periods (Fig. 2–4).

For $NO_x$, $NO_2$, and $PM_{10}$, the trends are based on the monthly average concentrations, and for $O_3$, they are based on rolling eight-hour daily maximum concentrations. The reason for this division is that the relative risk of $O_3$, used in our health-impact

calculations, is based on eight-hour daily maximum values. In Fig. 2–4, the blue lines with blue rings represent all the measured data points. The red lines represent the median slope in µg m$^{-3}$ year$^{-1}$, calculated according to the Theil-Sen method (Theil, 1992; Sen, 1968). The green texts in top of the figures represent the values of the median slopes, and in brackets, the 95 % confidence intervals of the trends. The stars rightmost in the green texts represent the significance level of the trend-lines, where three stars means that $p < 0.001$, one star means that $0.01 < p < 0.05$, and no star means that $p > 0.05$, which implies

that the trend-line is not statistically significant within a 95 % CI.



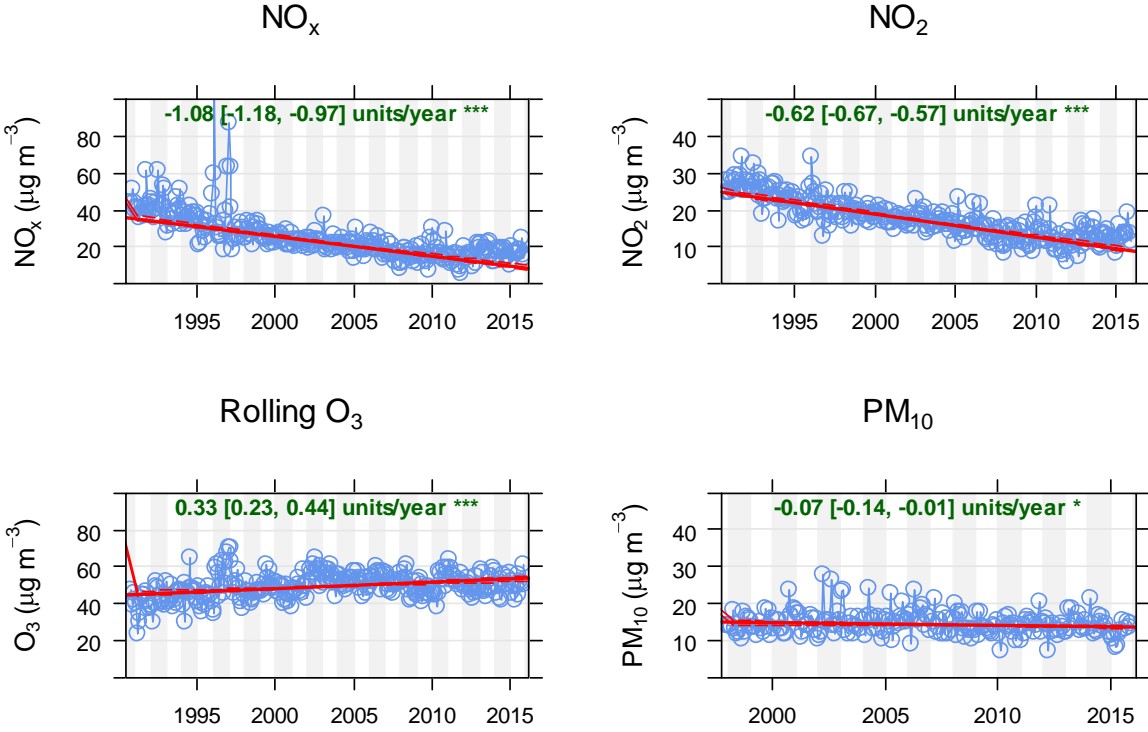

**Figure 2. Stockholm:** Trends in $NO_x$, $NO_2$, $O_3$, and $PM_{10}$, measured from 1990–2015. For $PM_{10}$, data are only for the period 1997–2015. The blue rings are monthly averages, and the calculated deseasonalised trends using the Theil-Sen method are shown as the red thick lines. Unit for the trends is µg m$^{-3}$ year$^{-1}$, and the values in parentheses are 95 % confidence intervals.





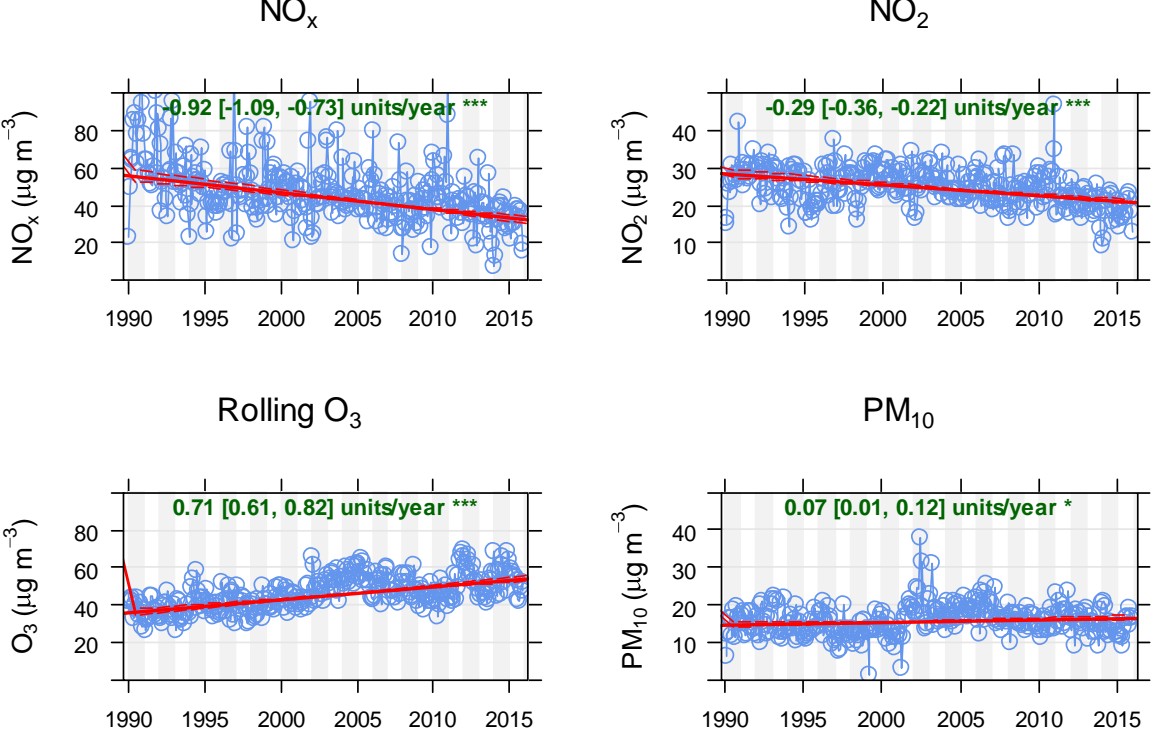

**Figure 3. Gothenburg:** Trends in $NO_x$, $NO_2$, $O_3$, and $PM_{10}$, measured from 1990–2015. The blue rings are monthly averages, and the calculated deseasonalised trends using the Theil-Sen method are shown as the red thick lines. Unit for the trends is µg $m^{-3}$ $year^{-1}$, and the values in parentheses are 95 % confidence intervals.

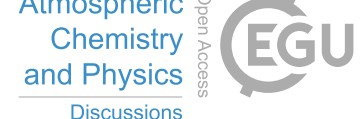



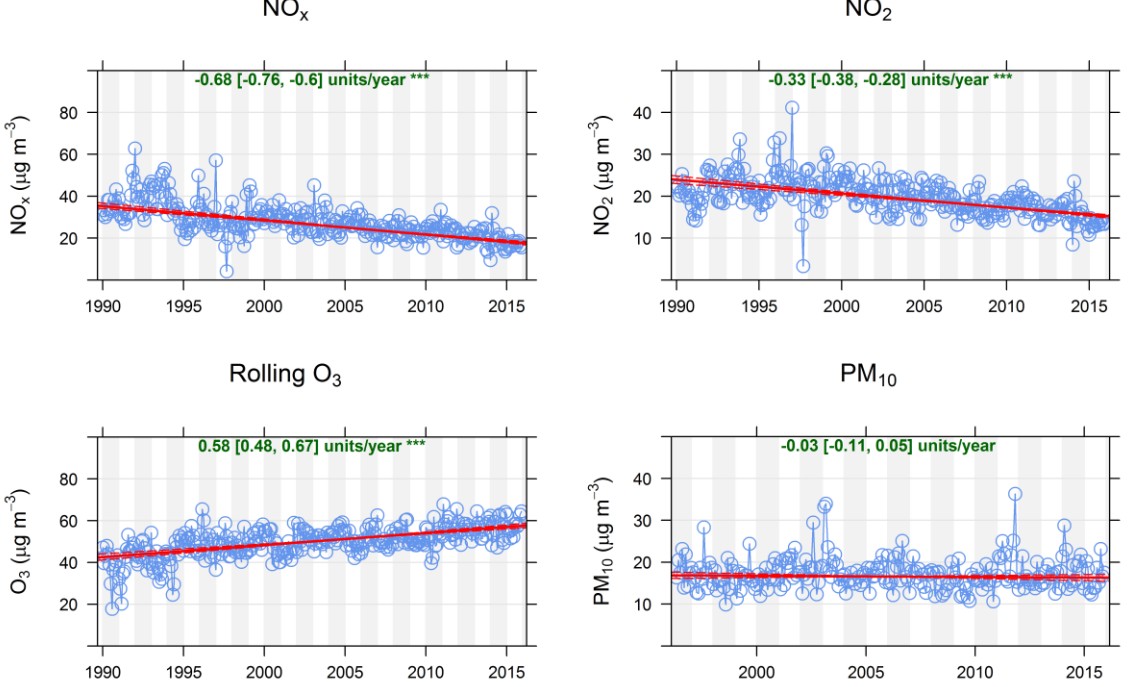

**Figure 4. Malmo:** Trends in $NO_x$, $NO_2$, $O_3$, and $PM_{10}$, measured from 1990–2015. For $PM_{10}$, data are only for the period 1996–2015. The blue rings are monthly averages, and the calculated deseasonalised trends using the Theil-Sen method are shown as the red thick lines. Unit for the trends is $\mu g\ m^{-3}\ year^{-1}$, and the values in parentheses are 95 % confidence intervals.

In Figure 5, we show the trends for the first period 1990–2007, and for the second period 2008–2015 separately (except for $PM_{10}$ which for Stockholm is from 1997–2007 and from 2008–2015, and for Malmo from 1996–2007 and from 2008–2015). In Stockholm, $NO_x$ and $NO_2$ concentrations have decreased significantly from 1990–2007, but for 2008–2015, there is even a tendency for increasing concentrations. The patterns are somewhat different in Gothenburg and Malmo.





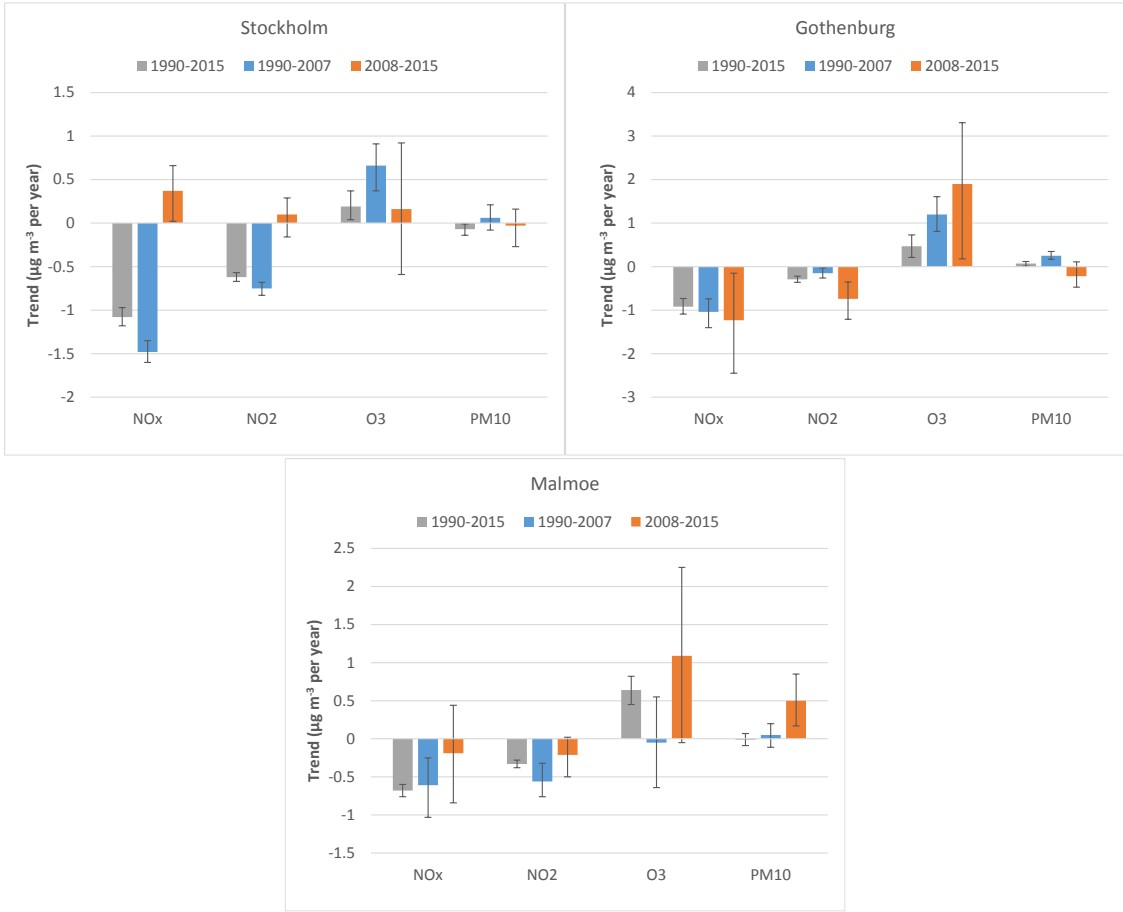

**Figure 5:** The trends for the measured pollutants in µg m$^{-3}$ year$^{-1}$ (95 % CI) for the whole period, and divided into two differnet periods.

There are also significantly different trends during weekdays and weekends in the three cities. In Fig. 6–8, the trends have been divided into weekdays and weekdays, reflecting the importance of local emissions compared to non-local emissions, where local emissions from traffic are more prominent during weekdays compared to weekends. Fig. 6–8 are otherwise designed in the same manner as Fig 2–4. For NO$_x$ and NO$_2$, the downward trends in all cities are more prominent during

10 weekdays compared to weekends, indicating that local emission reductions, mainly from traffic, have had the greatest impact. For O$_3$, the increasing trends in Gothenburg and Malmo are slightly higher during the weekends compared to the weekdays, while this does not apply to Stockholm. The trends related to PM$_{10}$ are less clear, and they are also in many cases not statistically significant. Possible reasons for the appearances of the trends are further analysed in the discussion section.





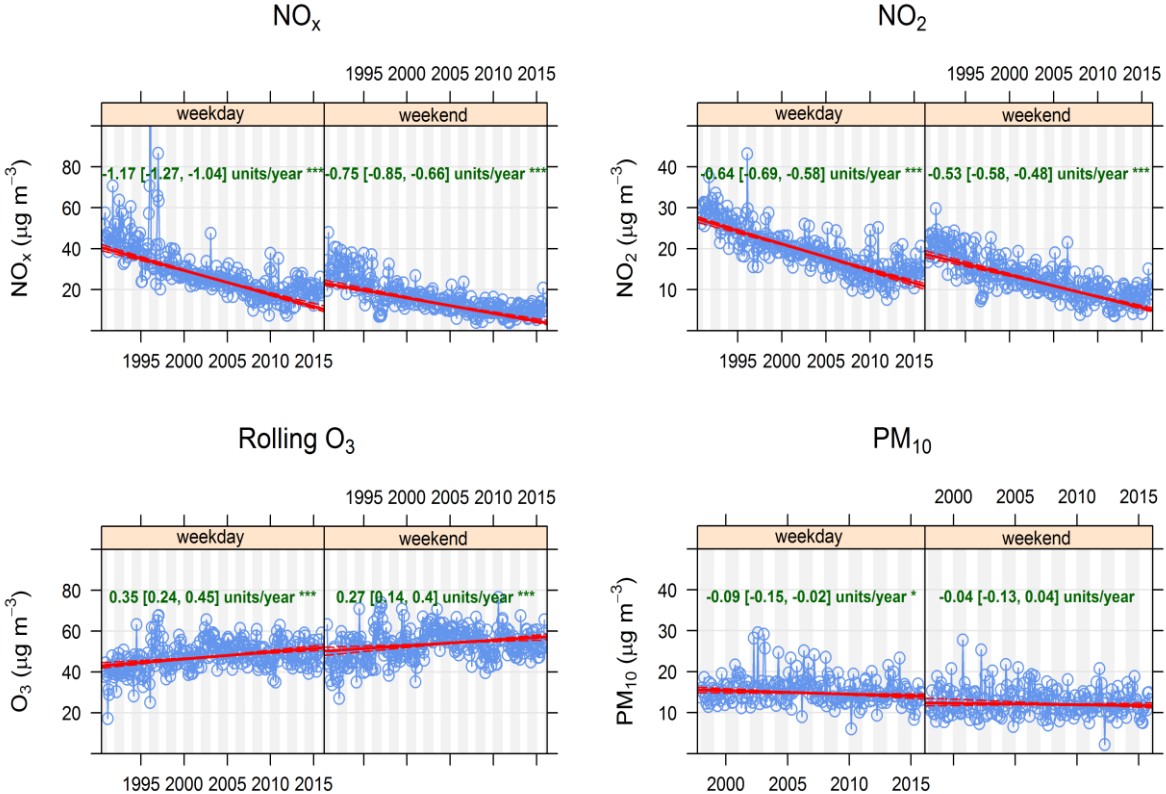

**Figure 6. Stockholm:** The trends for $NO_x$, $NO_2$, $O_3$, and $PM_{10}$, divided into weekdays and weekends.



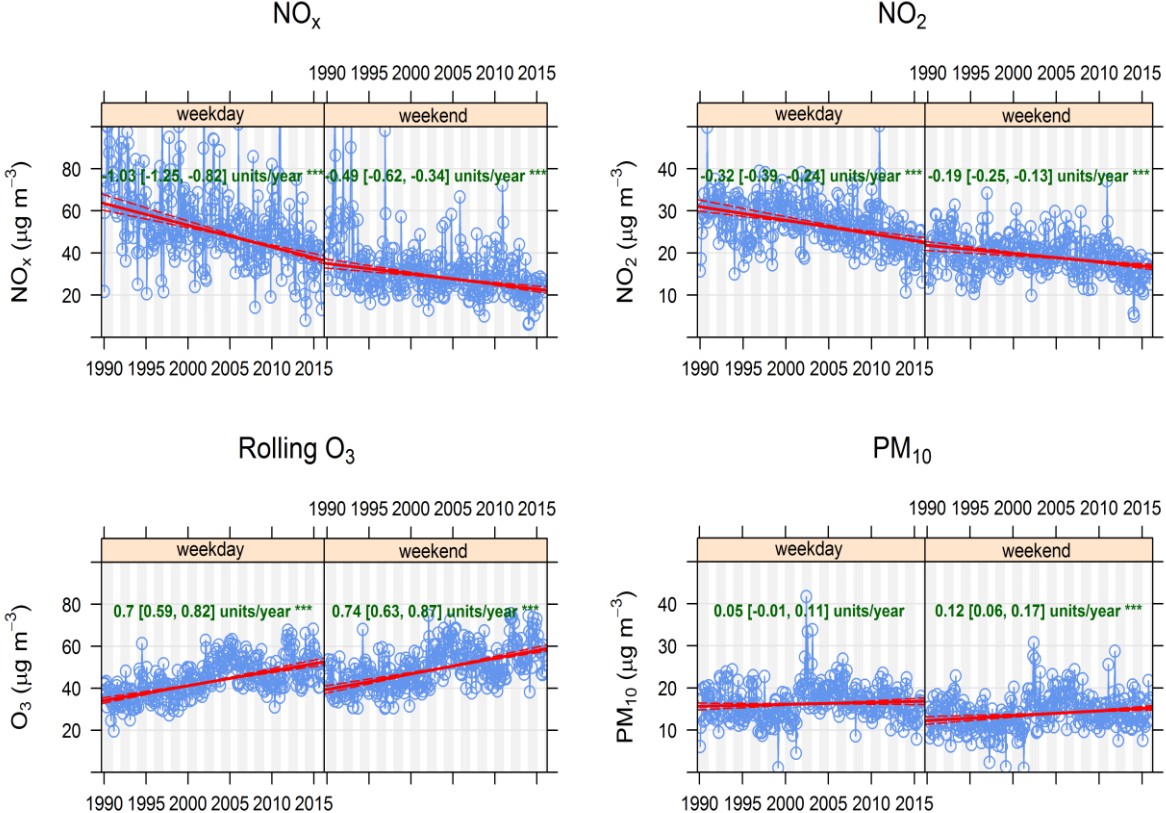

**Figure 7. Gothenburg:** The trends for $NO_x$, $NO_2$, $O_3$, and $PM_{10}$, divided into weekdays and weekends.




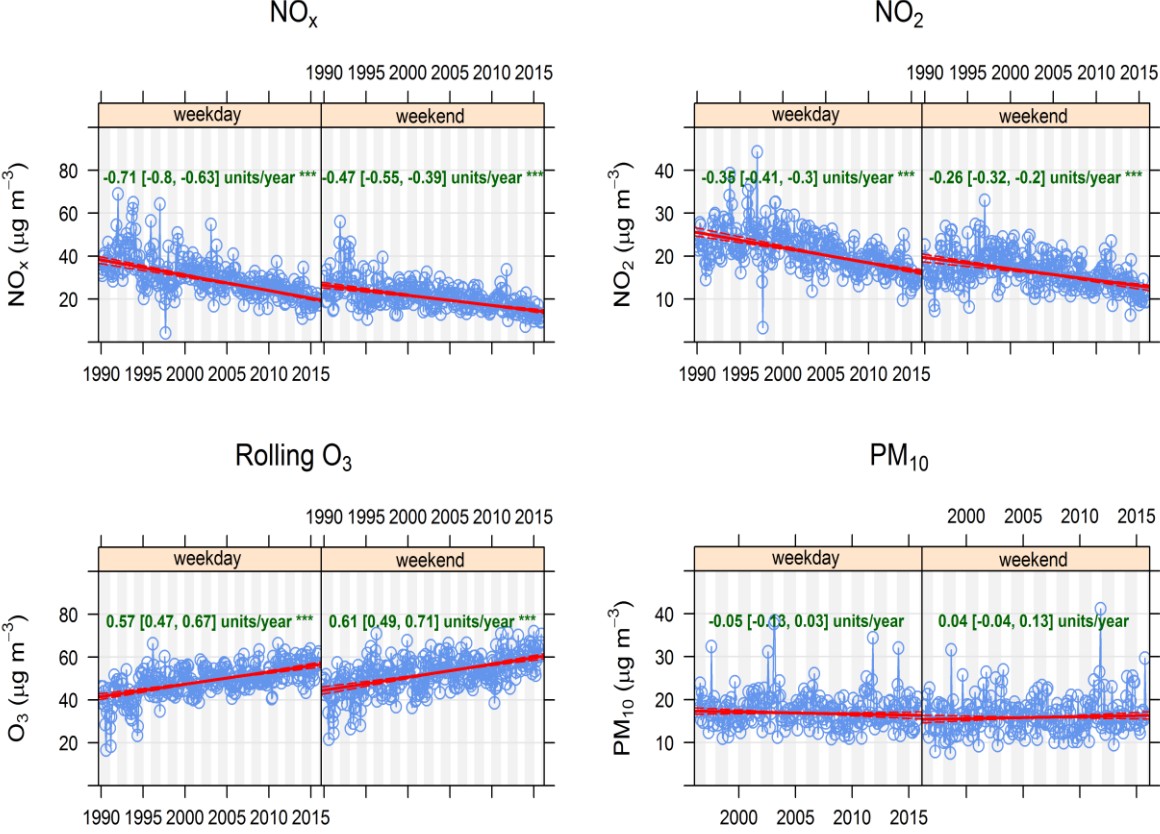

**Figure 8. Malmo:** The trends for $NO_x$, $NO_2$, $O_3$, and $PM_{10}$, divided into weekdays and weekends.

Table 1 summarizes the deseasonalised trends in concentration for $NO_x$, $NO_2$, $O_3$, and $PM_{10}$ in Stockholm, Gothenburg, and Malmo, presented in Fig. 2–4. Here we have used the median changes during the period, without considering the differences in different parts of the period (Fig 5). All trends, except for $PM_{10}$ in Malmo, are statistically significant within a 95 % CI. It can be noted that the rate of decreasing $NO_x$ is highest in Stockholm, and then comes Gothenburg and last Malmo, whereas the opposite order is for the rate of increasing $O_3$ concentrations. If the increase in $O_3$ was only associated with local NO titration, the opposite would be expected, so this indicates that other factors are also important for the $O_3$ trends, like the $O_3$ concentrations at the regional background sites outside the cities.



**Table 1:** The median change in µg m$^{-3}$ year$^{-1}$, and in brackets, the 95 % CI change per year in pollutant concentrations at the measuring stations in Stockholm, Gothenburg, and Malmo, according to the given time periods. Note that the trend of PM$_{10}$ in Malmo is not statistically significant.

| City | NO$_x$; time period | NO$_2$; time period | O$_3$; time period (8-hour daily maximum) | PM$_{10}$; time period |
|---|---|---|---|---|
| Stockholm; urban background | - 1.08 (- 1.18; - 0.97) *1990–2015* | - 0.62 (- 0.67; - 0.57) *1990–2015* | 0.33 (0.23; 0.44) *1990–2015* | - 0.07 (- 0.14; - 0.01) *1997–2015* |
| Gothenburg; urban background | - 0.92 (- 1.09; - 0.73) *1990–2015* | - 0.29 (- 0.36; - 0.22) *1990–2015* | 0.71 (0.61; 0.82) *1990–2015* | 0.07 (0.01; 0.12) *1990–2015* |
| Malmo; urban background | - 0.68 (- 0.76; - 0.60) *1990–2015* | - 0.33 (- 0.38; - 0.28) *1990–2015* | 0.58 (0.48; 0.67) *1990–2015* | - 0.03 (- 0.11; 0.05) *1996–2015* |

## 3.2 Health impact assessment associated with the trends in air pollution concentrations

### 3.2.1 Change in life expectancy

In order to estimate the health impacts associated with changing concentrations of NO$_x$, NO$_2$, O$_3$, and PM$_{10}$ in Stockholm Gothenburg, and Malmo, the concentration changes during the given time periods for the pollutants, presented in Table 1, and the relative risks, presented in section 2.4, have been used. Adjustments for population-weighted exposure concentrations have also been performed in the cases where data are available (Fig. A1–A4). The changes in life expectancy are calculated with the mean values and the 95 % confidence intervals of the relative risks, while for the trends and the population-weighted exposure concentrations, only the median and the mean values, respectively, have been used, but without considering their confidence intervals. NO$_x$ and NO$_2$ exhibit decreasing trends in all cities, which means an increase in life expectancy (positive values). The opposite applies to O$_3$, with increasing trends in all cities. For PM$_{10}$, there is a decreasing trend in Stockholm, with an increase in life expectancy, and an increasing trend in Gothenburg, with a decrease in life expectancy. For PM$_{10}$ in Malmo, however, no life expectancy change has been possible to calculate, due to the lack of a significant trend (see Fig. 4). The detailed results are presented in Table 2.

The largest increase in life expectancy is expected due to the reduction in NO$_x$, and somewhat less due to the reduction in NO$_2$ concentrations. The decrease in NO$_x$ levels corresponds to up to about 20 % of the total population life expectancy increase of between 54 and 70 months during the period 1990–2015 (Table A1.). In opposite, the increase in ozone exposure might have caused the life expectancy to decrease with one to two months. For PM$_{10}$, the effects are very mixed; in Stockholm, there might have been a small increase in life expectancy, whereas in Gothenburg, a small decrease has appeared, and for Malmo, no life expectancy change has been possible to calculate.



**Table 2:** Change in life expectancy in months with 95 % CI in brackets, caused by change in exposure during the measured periods. Decreasing trends are associated with an increase in life expectancy, and increasing trends are associated with a decrease in life expectancy (minus signs). The change in life expectancy, adjusted for population-weighted exposure concentrations (see Section 2.5 and Fig. A1–A4), are presented in bold below the ordinary values. Note that the trend of $PM_{10}$ in Stockholm is only for the period 1997–2015.

| Pollutant | Stockholm | Gothenburg | Malmo |
|---|---|---|---|
| $NO_x$ | 21 (11–32) | 17 (9–26) | 13 (7–19) |
| | **10 (5–15)** | **11 (5–16)** | |
| $NO_2$ | 13 (6–21) | 6 (3–9) | 7 (3–11) |
| $O_3$ | -1 (-0.5 – -2) | -2 (-1 – -4) | -2 (-1 – -4) |
| | **-1 (-0.6 – -3)** | | |
| $PM_{10}$ | 0.6 (0–1) | -1 (0 – -2) | No significant change |
| | **0.2 (0.0–0.5)** | | |

## 4 Discussion

### 4.1 Trends in concentrations and explanatory factors

In all studied cities, the $NO_x$ and $NO_2$ trends for urban background concentrations are descending. For Stockholm, the downward trends decrease and diminish almost entirely from 2008 onwards, and exhibit rather an increasing trend (Fig. 5). An increased share of diesel vehicles, that have taken place during this time period, can at least partially explain these trend changes (Krecl et al., 2017). However, despite an increased share of diesel vehicles in Sweden during this period, these trend changes during the latter part of the measurement periods are not apparent in Gothenburg and Malmo. The reason for this is not known, but could be related to the placement of the measuring station, and/or the local traffic situation in the vicinity. The descending trends for $NO_x$ and $NO_2$ in Stockholm are most evident during the period 1990–2000. Vigorously reduced emissions from road traffic, due to requirements of catalytic converters on new cars from the 1989 models, have largely contributed to these negative trends. The continuing reductions are also caused by stricter emission standards for new vehicles, congestion charges, and a greater proportion of clean vehicles within the city (Hurkmans et al., 2017; SLB, 2015). Comparing $NO_x$ and $NO_2$ in the three cities, the decreasing trends for $NO_x$ are much more pronounced compared to $NO_2$. The proportion of $NO_x$ that is $NO_2$ has been shown to be increasing at kerbside sites in many cities, and this has partly been attributed to more directly emitted $NO_2$ due to the increase in diesel vehicles equipped with oxidation catalysts (Carslaw, 2005) and particle filters (Grice et al., 2009). For urban background sites, the observed increased share of $NO_2$, and associated slower decline in concentrations compared to $NO_x$, are mainly related to the change in $O_3/NO_x$ equilibrium, caused by decreased $NO_x$ and increased $O_3$ concentrations (Keuken et al., 2009).

The $O_3$ levels in Stockholm (Fig. 2 and 6) exhibit increasing trends during the period. More stringent emission standards with reduced emissions of NO mean that less $O_3$ is consumed, due to a reduction of the NO titration, and thereby arises an increasing trend, as seen in many other cities in Europe (Sicard et al., 2016).



PM$_{10}$ exhibits decreasing trends in Stockholm (Fig. 2 and 6), however, not significant during weekends. PM$_{10}$ is a mixture of locally produced particles and long-range transport of particles from emissions in other countries. The decreasing trend in urban background can be explained by the reduced use of studded tires, and also by dust binding that has been introduced in the recent years (SLB, 2015).

In Gothenburg, NO$_x$ and NO$_2$ (Fig. 3 and 7) exhibit decreasing trends, largely because of the same reasons as in Stockholm, but the total concentration levels are somewhat higher compared to Stockholm and Malmo. Gothenburg is a traffic-intensive city, with major traffic routes passing along the low-lying valleys, which possibly can explain these higher NO$_x$ and NO$_2$ concentrations compared to Stockholm and Malmo (Gothenburg City, 2015). For O$_3$, there are like for Stockholm increasing trends. Unlike Stockholm, Gothenburg exhibits increasing trends for PM$_{10}$ during the measurement period (Fig. 3 and 7), however, not significant during weekdays. According to Fig. 5, the trend for PM$_{10}$ in Gothenburg from 2008 onwards is negative, but not significant. During 2006, measures were established in order to reduce the levels of PM$_{10}$. Dissemination of dust-binding agents, prohibiting the use of studded tires, and the introduction of congestion charges during 2013 may have contributed to this changing trend (Gothenburg City, 2015; Gothenburg Annual Report, 2015).

In Malmo, there are significant downward trends for NO$_x$ and NO$_2$ (Fig. 4 and 8). Through the inclusion of mitigation
strategies regarding NO$_x$ emissions, several traffic-reducing measures have been implemented in the central part of Malmo, and this is estimated to have caused a 17 % traffic reduction (Malmo, 2015). This reduction may at least partially explain these downward trends. The urban background levels of O$_3$ exhibit increasing trends in Malmo (Fig. 4 and 8). This trend has been observed since the O$_3$ measurements began in the end of the 1980´s (Malmo, 2015). For PM$_{10}$, the trends are not significant, neither for weekdays, weekends, or for the entire period (Fig. 4 and 8).

For O$_3$ and NO$_x$, there are more or less clear anti-correlations between these two pollutants in all the three cities, to a large extent caused by the titration effect, where the loss of O$_3$ due to reaction with NO is larger than the photochemical production of O$_3$. However, when comparing the O$_3$ and NO$_x$ trends in three cities, the most rapidly increasing trend of O$_3$ do not correspond to the most rapidly decreasing trend of NO$_x$. This would be expected if there are only the photochemical reactions that determine the observed increases in O$_3$. The increasing trend of O$_3$ in Stockholm, based on 8-hour daily maximum values,
is weaker compared to Gothenburg and Malmo, even though the NO$_x$ trend in Stockholm exhibits a sharper decline compared to Gothenburg and Malmo. This may, however, be explained by considering the trends in the regional background concentrations of O$_3$. The regional background concentrations of O$_3$ at Aspvreten outside of Stockholm shows a more sharply decreasing trend of O$_3$ compared to the regional background sites outside of Gothenburg and Malmo (SEPA, 2017a).

As mentioned in section 3.1, large differences between the trends during weekdays and weekends indicate that local
emissions have had the greatest impact. As shown in Fig. 6–8, the differences between weekends and weekdays are most prominent for NO$_x$ and NO$_2$. Based on annual means, the measuring stations that measure regional background air outside of the three cities exhibit decreasing trends for NO$_2$ during the period 1990–2015 (SEPA, 2017b). The decreasing trends for NO$_2$ at the urban background sites (Fig. 2–4), based on monthly averages, exhibit much sharper declines compared to the trends at the regional background sites, indicating that the these trends are mainly caused by local emission reductions.



For PM$_{10}$, the concentrations measured at the regional background sites during the period 1990–2015 are inadequate when it comes to assessing their impact on the urban background concentrations. For Stockholm (Aspvreten), the trend is certainly decreasing during the period, but it is fragmented with lack of data during parts of the period. For Gothenburg (Råö) and Malmo (Vavihill), there are only data for very limited parts of the period, and no conclusions about their impact on the urban

background concentrations are possible to do (SEPA, 2017c).

For NO$_x$, no continuous data measured at the regional background sites are available for the period 1990–2015. However, the NO$_x$ concentrations measured in 2016 at Norr Malma (a regional background site north of Stockholm) show very low values in comparison with the concentrations measured at the urban background site in Stockholm (SLB, 2017). Similar relationships can be assumed to apply also for the entire measurement period (1990–2015). Consequently, the NO$_x$ trends

measured at the urban background sites (Fig. 2–4) are not affected by the NO$_x$ concentrations at the regional background sites. The clear difference between the trends measured during weekdays and weekends (Fig. 6–8) provides further support for local emissions reductions being the most important factor.

## 4.2 Comparisons between our trends and the trends in U.S. and Europe as a whole

Regarding PM$_{10}$, the trends in the Swedish cities do not exhibit the same decrease as the trends in the U.S. or in Europe as a

whole. An important reason for this is the large contribution of mechanically generated road-dust particles in the Swedish cities. In Stockholm, up to 90 % of the mass fraction of PM$_{10}$ is generated from road abrasion, which is mostly caused by the use of studded tires during winter-time, and the concentration trend for PM$_{10}$ is not significantly influenced by the reduction of exhaust emissions (Johansson et al., 2007). Similar conditions regarding the composition of PM$_{10}$ prevail in Gothenburg (Grundström et al., 2015). In Europe as a whole, the PM$_{10}$ concentrations exhibit a decreasing trend during the period 1990–

2010, and this is mainly caused by reduced emissions of both primary PM and precursors of secondary PM within the European countries (EEA, 2017).

According to Geddes et al. (2016), where the global population-weighted annual mean concentrations of NO$_2$ from 1996– 2012 were analysed, the average downward trend in North America (Canada and U.S.) is 4.7 % per year, which is the most heavily declining trend compared to all the other regions in the world. For Western Europe, the average downward trend per

year is, according to Geddes et al. (2016), 2.5 % per year. This downward trend is in the same magnitude as the NO$_2$ trends in Stockholm, Gothenburg, and Malmo (Fig. 2, 3, and 4).

For O$_3$, the comparison is a little bit more complicated, since the trends may differ depending on the measure that is used. In the U.K., during the period 1993–2011, the mean and median concentration trends were positive, while the maximum concentration trend was negative (Munir et al., 2013). In Europe as a whole, the O$_3$ concentrations are influenced by both local

emissions, intercontinental inflow, and the meteorological conditions. The O$_3$ concentrations, and their changes in Europe during the period 1990–2010, are very different depending on time and place, and no unambiguous explanation or interpretation is possible to do (EEA, 2017). Considering the O$_3$ trends in Fig. 2–4, where these are based on 8-hour daily maximum values, all the trends are increasing. However, the O$_3$ concentrations associated with monthly averages also show



increasing trends for all the three cities during 1990–2015, and as mentioned in section 4.1, the increasing $O_3$ concentrations in the Swedish cities are probably largely caused by a reduced titration effect, which in turn is caused by decreasing NO concentrations.

### 4.3 Impacts on life expectancy

In general, the impact of $NO_x$ and $NO_2$ on life expectancy among the populations is much larger compared to the impact of $O_3$ and $PM_{10}$. The calculated gain in life expectancy, associated with decreasing $NO_x$ trends (Table 2), contributes up to as much as about 20 % of the total gain in life expectancy during the period 1990–2015 (Table A1). However, since the $O_3$ concentrations exhibit increasing trends during the same period, and thereby giving rise to a loss in life expectancy, the summarized effects of $NO_x$ exposure and $O_3$ exposure may be relevant to consider. For Stockholm, where population-weighted

exposure concentrations are available for both $NO_x$ and $O_3$, the mean value of the gain in life expectancy for $NO_x$, which is ten months, should be summed up with the mean value of the loss of life expectancy for $O_3$, which is one month, resulting in a net gain of 9 months. The very small corresponding gain in life expectancy of 0.2 months, associated with exposure to $PM_{10}$, can also be taken into account. However, the relatively weak trends associated with $PM_{10}$, with a significance level of $0.01 < p < 0.05$ for Stockholm and Gothenburg, and a corresponding level of $p > 0.05$ for Malmo, can be compared with the trends for

the others pollutants, which in all cases exhibit significance levels of $p < 0.001$.

The large impact on life expectancy, which both $NO_x$ and $NO_2$ represent, makes these pollutants extra important to consider. $NO_x$ and $NO_2$ are both indicators of combustion-related air pollutants, and they may therefore be good indicators of population exposure to combustion-related air pollutants in general. Additionally, the change in life expectancy can be regarded as the tip of the ice-berg, since reductions in $NO_x$ concentrations are also expected to cause reductions in cardiovascular and respiratory

morbidity (Johansson et al., 2009).

Health impact assessments from other studies clearly suggest that public health could largely benefit from better air quality (e.g. Künzli, 2002). However, estimates of health benefits of reducing air pollution are dependent on the indicator that is used, and on the shape of the concentration-response functions (Pope et al., 2015). In addition, Pope et al. (2015) also discusses recent evidence that the shape of the concentration-response function may be supra linear (dose-response relationship with a

negative second derivative) across wide ranges of exposure, as has been shown in the case of exposure to $PM_{2.5}$. It means that incremental pollution-abatement efforts may yield greater benefit in relatively clean areas compared to highly polluted areas. This motivates actions to be taken even in areas with relatively low levels of air pollutants.

### 4.4 Uncertainties associated with the measurements and the relative risks

In general, the results are highly sensitive to the relative risks obtained from previous epidemiological studies. For $NO_x$, we

use the RR 1.06 (95 % CI 1.03–1.09) per 10 µg m$^{-3}$ increase, based on a recent cohort study in Gothenburg (Stockfelt et al., 2015). Considering the similarities between Gothenburg, Stockholm, and Malmo regarding climate, vehicle fleet, and type of city, this RR may be representative for all these three Swedish cities. The other option could have been to choose the RR 1.08



(95 % CI 1.06–1.11), which comes from on a previous long-term cohort study in Oslo (Nafstad et al., 2004), but we decided to implement the RR from the Swedish study, with exposure levels closer to the actual situation. For $NO_2$, we use the pooled RR estimate of 1.066 (95 % CI 1.029–1.104) per 10 µg m$^{-3}$ increase, from a meta-analysis (Faustini et al., 2014). The other option could have been to choose the RR 1.05 (95 % CI 1.03–1.08) from Hoek et al. (2013), which is also based on a meta-analysis from several studies in different countries and continents, but we considered the study group in Faustini et al. (2014) as more relevant for the Swedish situation, since this RR is based only on European studies.

When health impact assessments are based on either $NO_x$ or $NO_2$, there are some issues to consider regarding the difference between the two metrics. $NO_2$ represents a sub fraction of $NO_x$, and the difference between the two metrics is dependent on the content of NO in $NO_x$. Considering the graphs in Fig. 2–4, where the trends for $NO_x$ and $NO_2$ in the three cities are presented during the same time periods, it appears that $NO_x$ to the relatively large part consists of $NO_2$. Based on these linear trends, $NO_2$ comprises between approximately 60 and 80 % of the $NO_x$ content. To distinguish the health effects of NO and $NO_2$ may be difficult since the exposure usually occurs simultaneously. The relative risks associated with $NO_x$ and $NO_2$ in the previous subsection are, however, in the same magnitude. Since $NO_2$ is an indicator of combustion-related pollutants in general, it is not quite straightforward to distinguish its independent health impact from other pollutants which are correlated with $NO_2$. Especially difficult is it when it comes to long-term effects associated with exposure to $NO_2$, since the results from short-term studies do not apply, and toxicological evidence is moreover limited (REVIHAAP, 2013).

For $O_3$ exposure, the relative risk of 1.01 (95 % CI 1.005–1.02) per 10 µg m$^{-3}$ increase, that has been used in our calculations, is based on Turner et al. (2016), which is a large prospective long-term cohort study performed in the U.S. The other option could have been to choose the RR 1.014 (95 % CI 1.005–1.024) from Jerret et al. (2009), which is very close to the RR in Turner et al. 2016, but considering the time periods during which the studies have been conducted, Turner et al. (2016) is more in line with the trend analysis period in this study. The number of epidemiological studies focusing on long-term exposure to $O_3$ are relatively few, and no meta-analyses are available. This means that the RR values associated with $O_3$ exposure are based on relatively less observed materials compared to the RR values for $NO_2$ and $PM_{10}$, which are obtained from meta-analyses.

For $PM_{10}$, we use the RR value of 1.04 (95 % CI 1.00–1.09) per 10 µg m$^{-3}$ increase from Beelen et al. (2014), which is a meta-analysis based on 22 European cohorts. An important issue to consider in these Swedish cities is that the mass of $PM_{10}$ to a relatively small percentage consists of combustion-related particles. In the Swedish cities, non-exhaust traffic generated particles are the largest contributors to the mass of $PM_{10}$, which is clearly shown for Stockholm by Johansson et al. (2007), and for Gothenburg by Grundström et al. (2015). This means that $NO_x$ and $NO_2$, which both primarily indicate locally produced combustion-related traffic emissions, are in Sweden clearly separated from $PM_{10}$, which rather indicates non-exhaust emissions. From a health perspective, $PM_{10}$ is a reflective indicator of the effects of exposure to the coarse fraction of urban aerosols. The health effects related to exposure to the coarse fraction of particles differ from exposure to the fine (combustion related) fraction. Previous studies indicate an association between exposure to the coarse fraction of $PM_{10}$ and respiratory admissions, such as asthma and chronic obstructive pulmonary disease, while cardiovascular diseases are more closely linked to exposure to the fine fraction (Brunekreef and Forsberg, 2005).



### 4.5 Correlations between the different pollutants

Calculating the health effects associated with exposure to these different pollutants may be problematic, considering that exposure can occur simultaneously. When conducting health-impacts assessments, it is important to avoid double calculations when gain or loss in life expectancy are calculated as a result of decreasing or increasing trends. $NO_x$ and $NO_2$ are highly correlated, and the health impacts assessments, calculated as a gain in life expectancy, cannot be summed up, but they can rather be considered as different indicators of air pollutants, mainly originated from combustion processes. On the other hand, for these three Swedish cities, the correlation between $NO_x$ or $NO_2$ and $PM_{10}$ is very low, since the mass proportion of exhaust particles in $PM_{10}$ is very small (Johansson et al., 2007), and the health impacts associated with exposure to $NO_x$ and $PM_{10}$, respectively, can therefore largely be assumed to be independent of each other. Moreover, when considering simultaneous exposure to $NO_2$ and $PM_{2.5}$, the relative risks associated with these pollutants are not significantly affected when included in a two-pollutant model (Faustini et al., 2014).

The relative risks associated with $O_3$ exposure are not significantly affected when they are included in two-pollutant models. The increased risks of circulatory and respiratory mortality associated with exposure to $O_3$ remained stable after adjustment for $PM_{2.5}$ and $NO_2$ (Turner et al., 2016). Consequently, the health effects associated with exposure to $O_3$ can be assumed to be separated from the health effects from the other three pollutants, and the change in life expectancy can therefore be added or subtracted from the change in life expectancy associated with $NO_x$, $NO_2$, and $PM_{10}$.

### 4.6 The differences between urban-background concentrations and population-weighted exposure concentrations

As shown in Section 2.5 and Figure A1–A4, the average population-weighted exposure concentrations may differ substantially from the concentrations measured at the measuring stations. For $NO_x$ and $PM_{10}$, the average exposure concentrations for the urban population are lower compared to the measured urban-background concentrations. The reason for this is that the people who live in the outskirts of the urban area are exposed to lower concentrations compared to those living close to the measuring station in the central parts of the cities, where both exhaust emissions and the formation of mechanically generated particles are most apparent. For $O_3$, the average exposure concentrations are higher compared to the measured urban-background concentrations. This can be explained by the interaction between $NO_x$ and $O_3$, where the $O_3$ reaction with NO dominates over the formation of new $O_3$, and this reaction is more apparent in the central part of a city, where the NO concentration is relatively higher. This explains the higher average population-exposure to $O_3$ compared to the measured urban-background concentrations at a central monitoring station.

The population data within the cities are not constant over time, and the meteorological conditions may also vary over time. Consequently, to calculate a corresponding change in population-weighted exposure, based on an air pollution trend during 25 years, is connected with uncertainties. Linear regression between the concentrations at the measuring stations, and the corresponding population-weighted exposure concentrations for the available years (Fig. A1–A4), are the best estimates that



can be performed. A disadvantage is the limited number of years of calculated relationships, but where the $R^2$-values in the range of 0.7–0.9 may be considered as good.

Due to lack of data, the relations between urban-background concentrations and population-weighted exposure concentrations have not been possible to calculate for $NO_2$ in Stockholm, and not for $NO_2$, $O_3$ and $PM_{10}$ in Gothenburg, and
not for any of the pollutants in Malmo. Without these relations, the population-weighted exposure concentrations have to be based on the data obtained directly from the measuring stations. However, based on the already established relations (Fig. A1–A4), it can be assumed that the life expectancy values associated with $NO_x$ and $NO_2$ exposure, but without adjustment for population-weighted exposure, are overestimated, while the opposite applies to $O_3$ exposure.

## 5. Conclusions

The air pollution trends regarding $NO_x$, $NO_2$, and $O_3$ in Stockholm, Gothenburg, and Malmo exhibit statistically significance (95 % CI) tendencies during the measured time periods. However, for $PM_{10}$, the trends in Stockholm and Gothenburg are significant, but the trend in Malmo is non-significant. The slopes of the trends are very different, where the $NO_x$ and $NO_2$ concentrations in all cities exhibit decreasing trends, while $O_3$ in all cities exhibits increasing trends. For $PM_{10}$, the trends are less clear with a decreasing trend in Stockholm, an increasing trend in Gothenburg, and no significant trend in Malmo. When
the trends are divided into weekdays and weekends, it follows that the trends for $NO_x$ and $NO_2$ in all cities exhibit large differences, where the trends associated with weekdays exhibit sharper declines compared to the weekends. This phenomenon indicates that local emission reductions mostly explain those declines, since the traffic intensity is more prominent during weekdays. An anti-correlation between $NO_x$ and $O_3$ can be seen for all cities, which can be attributed to an increased or decreased titration effect, where NO has the capability of scavenging free oxygen atoms wherein $NO_2$ is formed. When the
trends in this article are compared to the trends in other studies, the trends associated with $NO_x$ and $NO_2$ are in the same magnitude as those measured in the U.S. and Europe as a whole, but the trends associated with $PM_{10}$ are completely different, mostly caused by the large contribution of mechanically generated particles in the Swedish cities.

The change in life expectancy, associated with the air pollution trends, is most obvious for $NO_x$ and $NO_2$, where up to about 20 % of the total increase in life expectancy during the period 1990–2015 can be attributed to decreasing $NO_x$ trends. Since
$NO_x$ and $NO_2$ are indicators of combustion-related air pollutants in general, an overall conclusion is that exposure to these air pollutants are particularly important in terms of health effects and health benefits. From a policy point of view, it is important to put resources in order to implement the most effective abatement strategies intended to decrease the air pollution concentrations, which apparently benefit the public health.

## 6. Acknowledgements

H. Orru's work was supported by the Estonian Ministry of Education and Research grant IUT34-17.



## 7. Competing interests

The authors declare that they have no conflict of interest.

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

**Appendix A**

**Table A1:** General facts about the three cities regarding population structure, life expectancy at birth, and baseline mortality in terms of the number of deaths per 100 000 inhabitants. The number of deaths per 100 000 inhabitants are age standardized according to the average population for the year 2000.

|  | Stockholm | | Gothenburg | | Malmo | |
|---|---|---|---|---|---|---|
|  | 1990 | 2015 | 1990 | 2015 | 1990 | 2015 |
| **Population size** | 1 040 907 | 1 515 017 | 465 474 | 572 799 | 223 663 | 301 706 |
| **Density of population (inhabitants per km²)** | 3 601 | 4 935 | 965 | 1 224 | 1 522 | 2 060 |
| **Expected life expectancy at birth** | 76.8 | 82.6 | 77.2 | 82.1 | 77.6 | 82.1 |
| **Baseline mortality (all causes, age standardized), number per 100 000 inhabitants.** | 1 157* | 833 | 1 187* | 862 | 1 162* | 849 |

**\* Extrapolated from a linear regression based on data from 1997–2015**



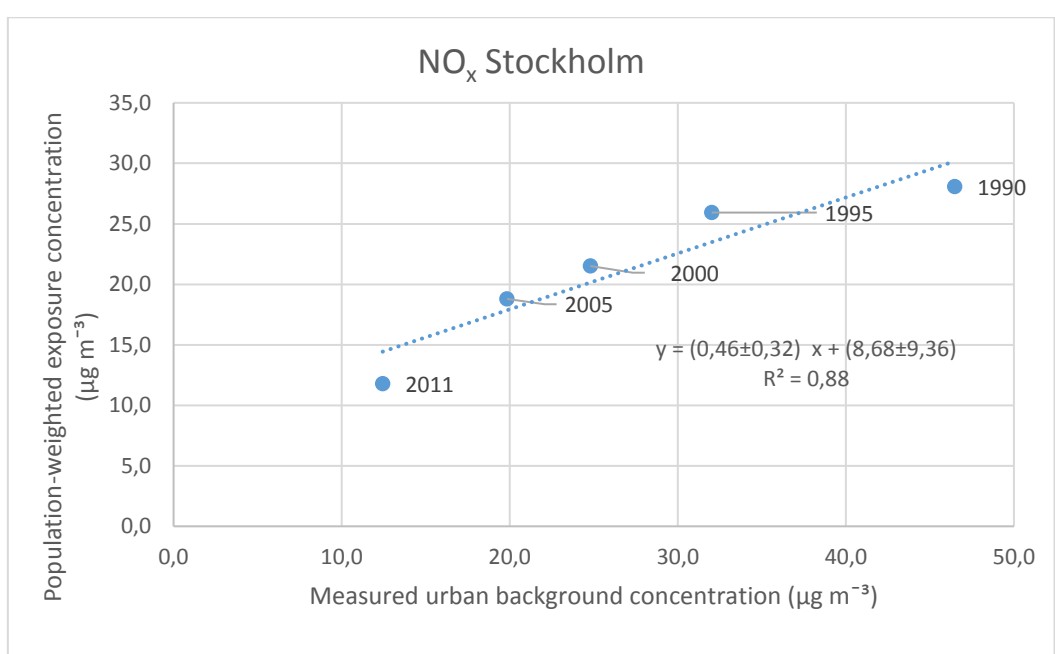

**Figure A1.** Relationship between measured urban-background concentrations and population-weighted exposure concentrations of $NO_x$ in Stockholm. Slope and intercept are specified with 95 % confidence intervals.

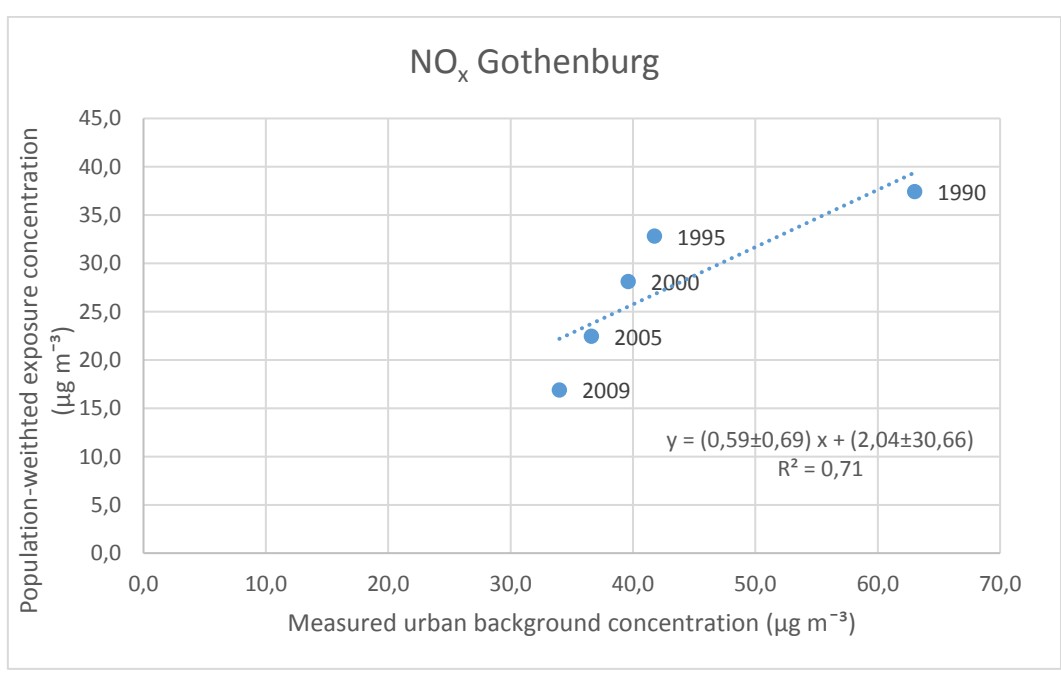

**Figure A2.** Relationship between measured urban-background concentrations and population-weighted exposure concentrations of $NO_x$ in Gothenburg. Slope and intercept are specified with 95 % confidence intervals.



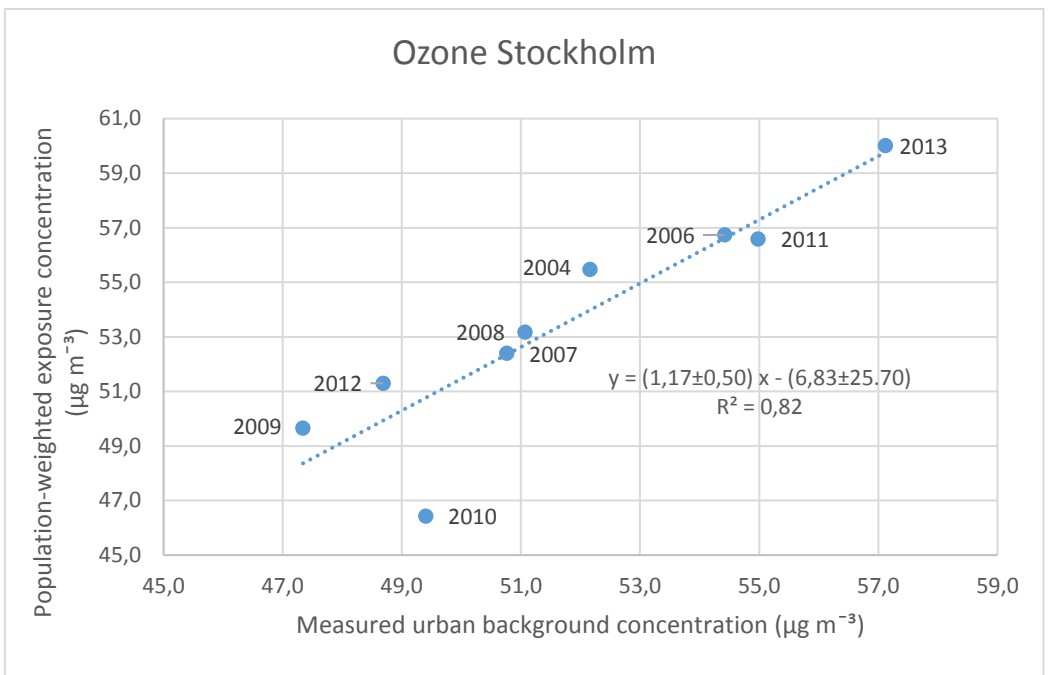

**Figure A3.** Relationship between measured urban-background concentrations and population-weighted exposure
concentrations of $O_3$ in Stockholm. Slope and intercept are specified with 95 % confidence intervals.



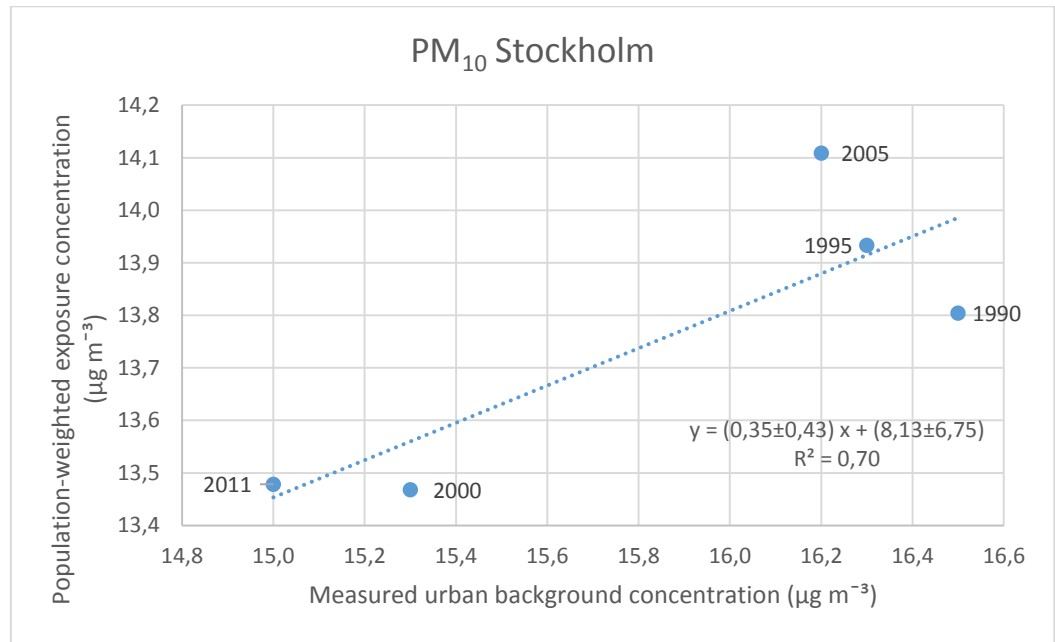

**Figure A4.** Relationship between measured urban-background concentrations and population-weighted exposure concentrations of PM$_{10}$ in Stockholm. Slope and intercept are specified with 95 % confidence intervals.

