# Peer review of "Trends in air pollutants and health impacts in three Swedish cities over the past three decades"

_Atmospheric Chemistry and Physics, 2018_

## Referee Comment (RC1) · E. Malmqvist (Referee) · 11 Apr 2018

This is a paper on the changing air pollutant levels over time and its health impacts. It is an important paper for countries yet having to work on reducing air pollutants to be able to see some quantified effects. However, some improvements and clarifications needs to be made. My main comment is that the paper is very long and results and methods could benefit from being presented in a more concise manner. The same for the discussion parts which also could be richer in references and less on speculation on own results. Below is some additional comments: Abbreviations should be spelled out first time used However, tests have shown that the calculations in life expectancy give very similar results regardless of the year (1997–2015) in which the population structure and mortality statistics are based on. Reference is needed for

this statement. We have applied relative risks obtained from previous epidemiological studies, where the relationships between mortality and exposure to NOx, NO2, O3, and PM10 have been analyzed. A discussion of choosing this over HRAPIE could be extended. We have done this for NOx in Stockholm and Gothenburg, and for O3 and PM10 in Stockholm. Why not for Malmö and why not for all places and not all places? Spatially resolved O3 concentrations in Stockholm are calculated from a combination of measurements and dispersion modelling of NOx concentrations. This could be a problem since you use same input data. Have the 03 model been validated by spatially distributed measurements? Some models have been validated with measurement data but to my knowledge not so much on data from earlier period. Please state year. It is quite unclear when measurements or modeling has been applied, please clarify. Please move some results and tables to supplemental material. The changes in life expectancy are calculated with 10 the mean values and the 95 % confidence intervals of the relative risks, while for the trends and the population-weighted exposure concentrations, only the median and the mean values, respectively, have been used, but without considering their confidence intervals. This belongs to methods Additionally, the particulate filters used for diesel vehicles will also give rise to an increased NO2/NOx ratio, since some of these filters work by oxidizing NO to NO2 (Grice et al., 2009; Wild et al., 2017). The O3 levels in Stockholm (Fig. 2 and 6) exhibit increasing trends during the period. More stringent emission standards with reduced emissions of NO mean that less O3 is consumed, due to a reduction of the NO titration, and thereby arises an 25 increasing trend. PM10 exhibits decreasing trends in Stockholm. Unclear if these two sentences match. O3 is also affected by the weather that year and this should be mentioned.

Please also note the supplement to this comment:
https://www.atmos-chem-phys-discuss.net/acp-2018-7/acp-2018-7-RC1-supplement.pdf

[Figure]

[Figure]

**Supplement:**

This is a paper on the changing air pollutant levels over time and its health impacts. It is an important paper for countries yet having to work on reducing air pollutants to be able to see some quantified effects. However, some improvements and clarifications needs to be made.

My main comment is that the paper is very long and results and methods could benefit from being presented in a more concise manner. The same for the discussion parts which also could be richer in references and less on speculation on own results. Below is some additional comments:

Abbreviations should be spelled out first time used

*However, tests have shown that the calculations in life expectancy give very similar results regardless of the year (1997–2015) in which the population structure and mortality statistics are based on.* Reference is needed for this statement.

We have applied relative risks obtained from previous epidemiological studies, where the relationships between mortality and exposure to $NO_x$, $NO_2$, $O_3$, and $PM_{10}$ have been analyzed.
A discussion of choosing this over HRAPIE could be extended.

We have done this for $NO_x$ in Stockholm and Gothenburg, and for $O_3$ and $PM_{10}$ in Stockholm. Why not for Malmö and why not for all places and not all places?

Spatially resolved $O_3$ concentrations in Stockholm are calculated from a combination of measurements and dispersion modelling of $NO_x$ concentrations. This could be a problem since you use same input data.

Have the O3 model been validated by spatially distributed measurements?

Some models have been validated with measurement data but to my knowledge not so much on data from earlier period. Please state year.

It is quite unclear when measurements or modeling has been applied, please clarify.

Please move some results and tables to supplemental material.

The changes in life expectancy are calculated with 10 the mean values and the 95 % confidence intervals of the relative risks, while for the trends and the population-weighted exposure concentrations, only the median and the mean values, respectively, have been used, but without considering their confidence intervals. This belongs to methods

Additionally, the particulate filters used for diesel vehicles will also give rise to an increased $NO_2/NO_x$ ratio, since some of these filters work by oxidizing NO to $NO_2$ (Grice et al., 2009; Wild et al., 2017).
The $O_3$ levels in Stockholm (Fig. 2 and 6) exhibit increasing trends during the period. More stringent emission standards with reduced emissions of NO mean that less $O_3$ is consumed, due to a reduction of the NO titration, and thereby arises an 25 increasing trend. $PM_{10}$ exhibits decreasing trends in Stockholm. Unclear if these two sentences match.

O3 is also affected by the weather that year and this should be mentioned.

---

## Author Comment (AC1) · 11 Apr 2018

Comments from Reviewer 2 and answers from the authors

First of all, we want to thank the reviewer for valuable comments regarding the manuscript. Below follow the comments and our answers. The changes have been implemented in the manuscript.

This is a paper on the changing air pollutant levels over time and its health impacts. It is an important paper for countries yet having to work on reducing air pollutants to be able to see some quantified effects. However, some improvements and clarifications needs to be made.

My main comment is that the paper is very long and results and methods could benefit

from being presented in a more concise manner. The same for the discussion parts which also could be richer in references and less on speculation on own results. Below is some additional comments:

Reply: We have shortened the discussion and added relevant references to make the discussion less speculative.

Abbreviations should be spelled out first time used:

Reply: The abbreviations PM2.5, PM10 and NOx have been changed so that they are defined the first time they are used.

However, tests have shown that the calculations in life expectancy give very similar results regardless of the year (1997–2015) in which the population structure and mortality statistics are based on. Reference is needed for this statement:

Reply: This statement is based on our own calculations of change in life expectancy by using the population structure and mortality statistics from different years. The calculated change in life expectancy differs by only a few days depending upon which year the population structure and the mortality statistics are based on. We have clarified that in the manuscript.

We have applied relative risks obtained from previous epidemiological studies, where the relationships between mortality and exposure to NOx, NO2, O3, and PM10 have been analyzed. A discussion of choosing this over HRAPIE could be extended.

Reply: We have extended this discussion in section 4.4. In HRAPIE, the RR associated with long-term exposure to NO2 is based on Hoek et al. (2013). In the Discussion Section 4.4, there is an explanation that we have chosen Faustini et al. (2014) instead of Hoek et al. (2013) since Faustini et al. (2014) also calculated this RR based on European studies only, which are more relevant for the Swedish conditions. For O3, our choice of the RR of 1.01 (95 % CI 1.005–1.02) per 10 $\mu$g m-3 increase in Turner et al. (2016) is very close to the RR of 1.014 (95 % CI 1.005–1.024) in Jerrett et al.

(2009), which is presented in HRAPIE. The study of Turner et al. (2016) has been implemented in a later period of time in comparison with Jerrett et al. (2009). This slightly later period of time is more in line with the period 1990–2015, which our trend analyzes are based on, making the RR in Turner et al. (2016) more relevant for our study. For PM10 and NOx, the HRAPIE report does not provide any RR for all-cause mortality associated with long-term exposure to these pollutants.

We have done this for NOx in Stockholm and Gothenburg, and for O3 and PM10 in Stockholm. Why not for Malmö and why not for all places and not all places?

Reply: We have added a sentence in the section "Method 2.5", stating that the lack of data is the reason for why this cannot be performed for all pollutants in all cities.

Spatially resolved O3 concentrations in Stockholm are calculated from a combination of measurements and dispersion modelling of NOx concentrations. This could be a problem since you use same input data.

Reply: This should not be a problem; the modelling of NOx has been validated with good results (Johansson et al., 2017; Molnar et al., 2015).

Have the 03 model been validated by spatially distributed measurements?

Reply: The ozone modelling has been validated based on measurements of ozone at four different sites in the Greater Stockholm area. These measurements cover the period 2003 to 2015, but with different amount of data for the different sites. R2 of model predictions versus measured monthly mean values for the 4 sites varied between 0.71 and 0.89, and the overall R2 (all data) was 0.77. We intend to send this for publication as soon as possible.

Some models have been validated with measurement data but to my knowledge not so much on data from earlier period. Please state year.

Reply: The model validations are based on several years, and not a specific year.

It is quite unclear when measurements or modeling has been applied, please clarify.

Reply: To keep it short, we refer to Johansson et al., 2017 (and references therein) for Stockholm, and for Gothenburg, we refer to Molnár et al., 2015.

Please move some results and tables to supplemental material.

Reply: We have moved Table 1 to Appendix A. Figure 6–8, where the trends are divided into weekdays and weekends, could possibly also be moved to the Appendix, but since they provide important info regarding the extent to which the emissions are local, we have chosen to have these figures in the result section.

The changes in life expectancy are calculated with 10 the mean values and the 95 % confidence intervals of the relative risks, while for the trends and the population-weighted exposure concentrations, only the median and the mean values, respectively, have been used, but without considering their confidence intervals. This belongs to methods:

Reply: We have added these sentences to the first paragraph of "Methods". For the sake of clarity, these sentences are repeated in the result section, where the change in life expectancy is described.

Additionally, the particulate filters used for diesel vehicles will also give rise to an increased NO2/NOx ratio, since some of these filters work by oxidizing NO to NO2 (Grice et al., 2009; Wild et al., 2017). The O3 levels in Stockholm (Fig. 2 and 6) exhibit increasing trends during the period. More stringent emission standards with reduced emissions of NO mean that less O3 is consumed, due to a reduction of the NO titration, and thereby arises an 25 increasing trend. PM10 exhibits decreasing trends in Stockholm. Unclear if these two sentences match.

Reply: We are not completely sure if we understand this comment correctly, but we have changed this text.

O3 is also affected by the weather that year and this should be mentioned.

[Figure]

Reply: Our trends are all based on measurements, and the relation between urban-background concentrations and population-weighted concentrations are taken from several years. The fluctuations between years driven by weather are not important for the long-term trend.

References

Faustini, A., Rapp, R., and Forastiere, F.: Nitrogen dioxide and mortality: review and meta-analysis of long-term studies, Eur. Respir. J., 44, 744-753. doi: 10.1183/09031936.00114713, 2014

Hoek, G., Krishnan, R. M., Beelen, R., Peters, A., Ostro, B., Brunekreef, B., and Kaufman, J. D.: Long-term air pollution exposure and cardio- respiratory mortality: a review, Environ. Health, 12, 43, doi: 10.1186/1476-069X-12-43, 2013

Jerrett, M., Burnett, R.T., Pope, C.A. 3rd, Ito, K., Thurston, G., Krewski, D., Shi, Y., Calle, E., and Thun, M.: Long-term ozone exposure and mortality, N. Engl. J. Med., 12, 1085-95, doi: 10.1056/NEJMoa0803894, 2009

Johansson, C., Löverheim, B., Schantz, P., Wahlgren, L., Almström, P., Markstedt, A., Strömgren, M., Forsberg, B., and Nilsson Sommar, J.: Impacts on air pollution and health by changing commuting from car to bicycle, Sci. Total. Environ., 584-585, 55-63, doi: 10.1016/j.scitotenv.2017.01.145, 2017

Molnar, P., Stockfelt, L., Barregard, L., and Sallsten, G.: Residential NOx exposure in a 35-year cohort study. Changes of exposure, and comparison with back extrapolation for historical exposure assessment, Atmos. Environ., 115, 62-69, doi: 10.1016/j.atmosenv.2015.05.055, 2015

Turner, M. C, Jerrett, M., Pope, C. A. 3rd, Krewski, D., Gapstur, S. M., Diver, W. R., Beckerman, B. S., Marshall, J. D., Su J., Crouse, D. L., and Burnett, R. T.: Long-Term Ozone Exposure and Mortality in a Large Prospective Study, Am. J. Respir. Crit. Care. Med., 193, 1134-1142, doi: 10.1164/rccm.201508-1633OC, 2016

---

## Referee Comment (RC2) · Anonymous Referee #3 · 10 Aug 2018

General comment: The manuscript reports trends for criteria air pollutants in three Swedish cities over the course of three decades. Reference relative risks on all-cause mortality are utilized in order to estimate changes in life expectancy. The intra-urban spatial variability is taken into account, in some cases, using dispersion modelling, for the determination of a population weighted exposure average. Although the paper doesn't introduce some methodological breakthrough, research results on the impacts that changing pollutant levels bear on life expectancy are not abundant in Europe. The manuscript would largely benefit from a more in-depth discussion on the factors that determine the observed trends, taking also into account regional inputs. Overall, atmospheric physical and chemical processes receive limited attention in the paper. The discussion and conclusions should focus more on policy implications for emission

control and public health, as should be expected by a manuscript on pollution trends. A substantial revision along these lines is necessary, incorporating also the following specific comments and edits.

Specific comments: Page 2, lines 20-22: While the decline of NOx emissions in the EU is larger as compared to PM, it is probably not as efficient as it has been initially projected (see the rates of attainment of national emission ceilings specified by the original NEC directive) and there is also a lot of between-country variation. Page 2, lines 23-30: The trend for O3 levels with regard to the target value for the protection of human health, during the period 1990-2014 in the EU, has been rather a decreasing one. This has to be taken into account and make a distinction between mean ozone levels and higher percentiles more relevant for short-term exposure. The importance of the O3 metric is already mentioned in line 28, but the authors should avoid indicating that O3 levels are increasing all over the board in Europe. Page 3: lines 14-21: It would be better to provide the fractional changes of premature mortality instead of the net numbers, especially since the populations to which the studies refer can't be adequately described in an introductory section. Page 4, lines 20-30: Indicate if these stations are regulatory monitoring stations which provide measurements according to the reference methods. Page 4, line 21: Indicate the sampling height in Malmo. Page 6, line 9: The value selected for NO2 differs from the one reported by Beelen et al. (2014) in the study which also provided the PM10 RR used here. This should be discussed in section 4.4. Figures 3-4: There are some extremely low mean monthly values for PM10 in Gothenburg and NOx -NO2 in Malmo. Provide an explanation, is it meteorology or else? Table 1: The Table repeats the information of Figures 2-4. It should be removed altogether. Section 4.1.: The section needs an overhaul. The discussion should be performed by pollutant at the first level and then city specific mentions should be made where important differences occur. More clarity is needed in the argumentation. The potential impact of regional emission reductions from non-transport sources should be incorporated in the discussion. Page 16, line 12: Correct the phrasing here. Also, it is not clear how the dieselization has led to the reduction of NOx emissions. Krecl et al.

[Figure]

(2017) report that NOx have remained constant during the process. Please elaborate. An indication of the change in the vehicle parc composition should be given in number. Page 16, lines 14-15: How does the location affect the trend? Indicating the distances from major roads could be informative, although the reported sampling heights are probably too large to represent direct road traffic emissions. Page 16, lines 20-23: Given the site types and the sampling heights, it is somewhat doubtful that the primary NO2 variability could be captured. Page 17, line 21: The whole discussion regarding the increasing ozone trend is obviously founded on the assumption that photochemical processes for ozone production in Sweden should be of minor importance. Otherwise the reduction of precursor emissions would generally lead to the long-term reduction of ozone as well, as it has been observed in various studies. Please, better clarify the dominant mechanism explaining the O3 presence in the urban setting. Page 17, lines 29-31: Given that PM10 has been associated with vehicular emissions, shouldn't a similar to NOx weekday-weekend pattern be observed? Justify this difference. Page 18, lines 7-10: The hypothesis for the whole period cannot be supported by just one year of data and moreover these regional background data should be better described. Also, it is not clear why there aren't long term regional background NOx data available, when at Page 17, line 32 the availability of such data for NO2 is stated. Page 20, line 24-34: The study which provided the RR for PM10 includes similar results on mortality associations for PMcoarse. This could be discussed.

Technical corrections: Page2, line 7: Check phrasing ("ending"). Page 2, line 11: Check phrasing, you could replace with ". . .of the apparent major health impact of exposure to air pollutants...". Page 2, line 14: Delete "amount of". Page, 2, line 16: Equals sign not in subscript. Page 4, line 13: . . .represents the urban background. Page 4, line 13-18: Remove the coordinate information. Page 5, line 21: Delete "decreasing". Page 6, line 22: Replace "increase" with "change" Page 7, lines 25-30: This information is already provided in the Figure caption. Remove accordingly from the text. Figure 5: Ensure that Malmo is spelled consistently throughout the manuscript. Page 14, lines 8-9: Check phrasing. It should be "If the change in O3 was only associated with local

NO titration. . .". Page 16, line 10: Decrease and diminish, pick one. Page 19, line 14: Can be compared or can't compare? Please rephrase the sentence. Page 19, line 16: Delete "extra". Page 20, line 12: "exposure occurs simultaneously"? What is meant here? Probably it refers to peak exposures. The same in section 4.5. Page 20, line 15: "Especially difficult is it. . .". Correct wording. Page 20, line 23: "Observed materials"? Page 21, line 28: You mean the population distribution within the cities? Page 22, line 29: NO2 is formed by the reaction of nitric oxide with O3. Figure A1-A4: Correct decimal separators.
* * *

---

## Author Comment (AC2) · 20 Sep 2018

First of all, we would like to thank the Reviewer for constructive comments and technical corrections! The changes in the manuscript have been made with track changes. The whole discussion section has been substantially revised, and large parts have been exchanged. Below follow the comments and our answers marked with yellow. Reviewers Comment: The discussion and conclusions should focus more on policy implications for emission as should be expected by a manuscript on pollution trends. Answer: We have expanded the Discussion with a new section 4.7, "Policy implications", and also added a sentence at the end of the section "Conclusions". Reviewers Comment: Page 2, lines 20-22: While the decline of NOx emissions in the EU is larger as compared to PM, it is probably not as efficient as it has been initially projected (see the rates of at-

tainment of national emission ceilings specified by the original NEC directive) and there is also a lot of between-country variation. Answer: We added a comment: It should also be noted that there are large between-country variations in NOx emission trends partly reflecting that some countries have had problems meeting the original National Emission Ceilings and the Air Quality directives (EEA, 2017). Reviewers Comment: Page 2, lines 23-30: The trend for O3 levels with regard to the target value for the protection of human health, during the period 1990-2014 in the EU, has been rather a decreasing one. This has to be taken into account and make a distinction between mean ozone levels and higher percentiles more relevant for short-term exposure. The importance of the O3 metric is already mentioned in line 28, but the authors should avoid indicating that O3 levels are increasing all over the board in Europe. Answer: We have added a comment: "Trends in ozone are also different for summer and winter, with mainly decreasing trends in summer and increasing in winter, and there are also some variations between cities in the EU (see EEA, 2016)." Reviewers Comment: Page 3: lines 14-21: It would be better to provide the fractional changes of premature mortality instead of the net numbers, especially since the populations to which the studies refer can't be adequately described in an introductory section. Answer: We agree that it would be beneficial to include the fractional changes. Unfortunately, since there are no information available regarding the fractional changes in the studies which have been referred to, we have kept the premature mortalities as net numbers instead of fractional changes. Reviewers Comment: Page 4, lines 20-30: Indicate if these stations are regulatory monitoring stations which provide measurements according to the reference methods. Answer: We added this info: They are all regulatory monitoring urban background stations using reference methods. Reviewers Comment: . Page 4, line 21: Indicate the sampling height in Malmo. Answer: We have added the sampling height of 20 m. Reviewers Comment: Page 6, line 9: The value selected for NO2 differs from the one reported by Beelen et al. (2014) in the study which also provided the PM10 RR used here. This should be discussed in section 4.4. Answer: The choice of RR in Faustini et al. (2014) instead of using Beelen et al. (2014) have been motivated in section

4.4. Reviewers Comment: Figures 3-4: There are some extremely low mean monthly values for PM10 in Gothenburg and NOx -NO2 in Malmo. Provide an explanation, is it meteorology or else? Answer: For NOx and NO2, we found some erroneous data that explained the low values. Therefore, Figure 4 has been replaced. The median trend is unaltered, while the upper and lower confidence interval only decrease by one hundredth $\mu$g m-3. For PM10, the low values are due to the way that the deseasonalisation is modelled. When we disregard the deseasonalisation, these extreme low values are not present. But we have decided to keep the deseasonalisation of the trend, in order to be consistent with all the other calculations. Reviewers Comment: Table 1: The Table repeats the information of Figures 2-4. It should be removed altogether. Answer: OK, Table removed.

Reviewers Comment: Section 4.1.: The section needs an overhaul. The discussion should be performed by pollutant at the first level and then city specific mentions should be made where important differences occur. More clarity is needed in the argumentation. The potential impact of regional emission reductions from non-transport sources should be incorporated in the discussion. Answer: Section 4.1 has been revised substantially. The discussion is based primarily on the pollutants and secondarily on the cities. The potential impact of regional emission reductions from non-transport sources has been incorporated Reviewers Comment: Page 16, line 12: Correct the phrasing here. Also, it is not clear how the dieselization has led to the reduction of NOx emissions. Krecl et al. (2017) report that NOx have remained constant during the process. Please elaborate. An indication of the change in the vehicle parc composition should be given in number. Answer: We have changed this and refer to statistics on diesel shares from BilSweden (2018). Reviewers Comment: Page 16, lines 14-15: How does the location affect the trend? Indicating the distances from major roads could be informative, although the reported sampling heights are probably too large to represent direct road traffic emissions. Answer: We agree, and we have removed this. Reviewers Comment: Page 16, lines 20-23: Given the site types and the sampling heights, it is somewhat doubtful that the primary NO2 variability could be captured. Answer:

[Figure]

Yes, we agree, and this is also clearly stated in the text in Section 4.1.1. Reviewers Comment: Page 17, line 21: The whole discussion regarding the increasing ozone trend is obviously founded on the assumption that photochemical processes for ozone production in Sweden should be of minor importance. Otherwise the reduction of precursor emissions would generally lead to the long-term reduction of ozone as well, as it has been observed in various studies. Please, better clarify the dominant mechanism explaining the O3 presence in the urban setting. Answer: The concentrations of ozone are lower at central urban background sites in the cities compared to outside the cities at rural background sites. This means that the net effect of the photochemistry involving ozone in the cities is that ozone is consumed, mainly due to the titration involving NOx.

Reviewers Comment Page 17, lines 29-31: Given that PM10 has been associated with vehicular emissions, shouldn't a similar to NOx weekday-weekend pattern be observed? Justify this difference. Answer: The main local source of PM10 is road wear and road dust suspension (clearly seen in Stockfelt et al., 2017 and Segersson et al., 2017). Since the emissions of road dust strongly depend on the wetness of the roads, as shown by Johansson et al. (2007) and Denby et al. (2013), the diurnal cycles will not follow the same pattern as vehicle exhaust from traffic. Reviewers Comment Page 18, lines 7-10: The hypothesis for the whole period cannot be supported by just one year of data and moreover these regional background data should be better described. Also, it is not clear why there aren't long term regional background NOx data available, when at Page 17, line 32 the availability of such data for NO2 is stated. Answer: This part is rewritten with a new Section 4.1.1. Local and non-local contributions are explained more detailed, underpinned with several new references. The regional background stations are described in Section 4.1.2. Reviewers Comment Page 20, line 24-34: The study which provided the RR for PM10 includes similar results on mortality associations for PMcoarse. This could be discussed. Answer: Since PM10 to a large extent consists of mechanically generated coarse particles in the Swedish cities, the similar results on mortality associated with PMcoarse in Beelen et al. (2014) provide increased support

for the RR that has been used. We have added a few sentences about this in Section 4.4, where RR associated with PM10 exposure is discussed. Technical corrections: Page2, line 7: Check phrasing ("ending"). The phrasing has been changed. Page 2, line 11: Check phrasing, you could replace with ". . .of the apparent major health impact of exposure to air pollutants...". It is changed to "associated with"

Page 2, line 14: Delete "amount of". Deleted Page, 2, line 16: Equals sign not in subscript. Changed Page 4, line 13: . . .represents the urban background. "the" urban is inserted. Page 4, line 13-18: Remove the coordinate information. Removed. Page 5, line 21: Delete "decreasing". Deleted. Page 6, line 22: Replace "increase" with "change" We don't want to change increase to "change", because it becomes illogical in relation to the following sentence, where the word decrease has been used. Page 7, lines 25-30: This information is already provided in the Figure caption. Remove accordingly from the text. Figure 5: Ensure that Malmo is spelled consistently throughout the manuscript. Lines 25-30 are removed, and the spelling Malmo is used consistently, including Figure 5. Page 14, lines 8-9: Check phrasing. It should be "If the change in O3 was only associated with local NO titration. . .". Increase is changed to "change". Page 16, line 10: Decrease and diminish, pick one. The whole section 4.1 has been changed. Page 19, line 14: Can be compared or can't compare? Please rephrase the sentence. Can is changed to "cannot". Page 19, line 16: Delete "extra". Deleted Page 20, line 12: "exposure occurs simultaneously"? What is meant here? Probably it refers to peak exposures. The same in section 4.5. It means that environmental exposures to NO2 and NO usually occur simultaneously, since the urban air contains both of these pollutants in varying proportions. Determining the effect of each pollutant can therefore be difficult. In Section 4.5 we clarify that double calculations regarding change in life expectancy occur if the effect of NOx and NO2 is summarized, but this is not the case for the others pollutants, where the effects are assumed to be independent of each other. Page 20, line 15: "Especially difficult is it. . .". Correct wording. This part has been removed Page 20, line 23: "Observed materials"? It is changed from "less observed materials" to "less amount of data" Page 21, line 28: You mean the population

distribution within the cities? Yes, "data" is changed to "distribution". Page 22, line 29: NO2 is formed by the reaction of nitric oxide with O3. Reformulated. Figure A1-A4: Correct decimal separators. The commas have been changed to point characters. References Beelen, R., Raaschou-Nielsen, O., Stafoggia, M., Andersen, Z. J., Weinmayr, G., Hoffmann, B., Wolf, K., Samoli, E., Fischer, P., Nieuwenhuijsen, M., Vineis, P., Xun, W. W., Katsouyanni, K., Dimakopoulou, K., Oudin, A., Forsberg, B., Modig, L., Havulinna, A. S., Lanki, T., Turunen, A., Oftedal, B., Nystad, W., Nafstad, P., De Faire, U., Pedersen, N. L., Östenson, C. G., Fratiglioni, L., Penell, J., Korek, M., Pershagen, G., Eriksen, K. T., Overvad, K., Ellermann, T., Eeftens, M., Peeters, P. H., Meliefste, K., Wang, M., Bueno-de-Mesquita, B., Sugiri, D., Krämer, U., Heinrich, J., de Hoogh, K., Key, T., Peters, A., Hampel, R., Concin, H., Nagel, G., Ineichen, A., Schaffner, E., Probst-Hensch, N., Künzli, N., Schindler, C., Schikowski, T., Adam, M., Phuleria, H., Vilier, A., Clavel-Chapelon, F., Declercq, C., Grioni, S., Krogh, V., Tsai, M. Y., Ricceri, F., Sacerdote, C., Galassi, C., Migliore, E., Ranzi, A., Cesaroni, G., Badaloni, C., Forastiere, F., Tamayo, I., Amiano, P., Dorronsoro, M., Katsoulis, M., Trichopoulou, A., Brunekreef, B., and Hoek, G.: Effects of long-term exposure to air pollution on natural-cause mortality: an analysis of 22 European cohorts within the multicentre ESCAPE project, Lancet, 383, 785-95, doi: 10.1016/S0140-6736(13)62158-3, 2014 BilSweden, 2018. http://www.bilsweden.se/statistik/arkiv-nyregistreringar_1 (2018-08-29) Denby, B.R., Sundvor I., Johansson C., Pirjola L., Ketzel M., Norman M., Kupiainen K. Gustafsson M., Blomqvist G., Omstedt G. 2013a. A coupled road dust surface moisture model to predict non-exhaust road traffic induced particle emissions (NOR-TRIP). Part 1: Road dust loading and suspension modelling. Elsivier, Atmospheric Environment 77, Volym 77, pp. 283-300. EEA, 2016. Air quality in Europe — 2016 report, No 28/2016. European Environment Agency. ISSN 1977-8449. EEA, 2017. Air quality in Europe — 2017 report, No 13/2017. European Environment Agency. ISSN 1725-9177 Faustini, A., Rapp, R., and Forastiere, F.: Nitrogen dioxide and mortality: review and meta-analysis of long-term studies, Eur. Respir. J., 44, 744-753. doi: 10.1183/09031936.00114713, 2014 Johansson, C., Norman, M., and Gidhagen,

L.: Spatial & temporal variations of PM10 and particle number concentrations in urban air, Environ. Monit. Assess., 127, 477-487, doi: 10.1007/s10661-0069296-4, 2007 Segersson, D., Eneroth, K., Gidhagen, L., Johansson, C., Omstedt, G., Engström Nylen, A., and Forsberg B.: Health Impact of PM10, PM2.5 and black carbon exposure due to different source sectors in Stockholm, Gothenburg and Umea, Sweden, Int. J. Environ. Res. Public. Health., 14, 742, doi: 10.3390/ijerph14070742, 2017 Stockfelt, L., Andersson, E.M., Molnár, P., Gidhagen, L., Segersson, D., Rosengren, A., Barregard, L., and Sallsten, G.: Long-term efiects of total and source-specific particulate air pollution on incident cardiovascular disease in Gothenburg, Sweden, Environ. Res., 158, 61-71, doi: 10.1016/j.envres.2017.05.036, 2017

---

## Author Response (AR1)

**First of all, we would like to thank the Reviewer for constructive comments and technical corrections!**

The changes in the manuscript have been made with track changes. The whole discussion section has been substantially revised, and large parts have been exchanged.

Below follow the comments and our answers marked with yellow.

Reviewers Comment: The discussion and conclusions should focus more on policy implications for emission as should be expected by a manuscript on pollution trends.

**Answer:**

We have expanded the Discussion with a new section 4.7, "Policy implications", and also added a sentence at the end of the section "Conclusions".

Reviewers Comment: Page 2, lines 20-22: While the decline of $NO_x$ emissions in the EU is larger as compared to PM, it is probably not as efficient as it has been initially projected (see the rates of attainment of national emission ceilings specified by the original NEC directive) and there is also a lot of between-country variation.

**Answer:**

We added a comment:

It should also be noted that there are large between-country variations in $NO_x$ emission trends partly reflecting that some countries have had problems meeting the original National Emission Ceilings and the Air Quality directives (EEA, 2017).

Reviewers Comment: Page 2, lines 23-30: The trend for O3 levels with regard to the target value for the protection of human health, during the period 1990-2014 in the EU, has been rather a decreasing one. This has to be taken into account and make a distinction between mean ozone levels and higher percentiles more relevant for short-term exposure. The importance of the $O_3$ metric is already mentioned in line 28, but the authors should avoid indicating that $O_3$ levels are increasing all over the board in Europe.

**Answer:**

We have added a comment:

"Trends in ozone are also different for summer and winter, with mainly decreasing trends in summer and increasing in winter, and there are also some variations between cities in the EU (see EEA, 2016)."

Reviewers Comment: Page 3: lines 14-21: It would be better to provide the fractional changes of premature mortality instead of the net numbers, especially since the populations to which the studies refer can't be adequately described in an introductory section.

**Answer:**

We agree that it would be beneficial to include the fractional changes. Unfortunately, since there are no information available regarding the fractional changes in the studies which have been referred to, we have kept the premature mortalities as net numbers instead of fractional changes.

Reviewers Comment: Page 4, lines 20-30: Indicate if these stations are regulatory monitoring stations which provide measurements according to the reference methods.

**Answer:**

We added this info:

They are all regulatory monitoring urban background stations using reference methods.

Reviewers Comment: . Page 4, line 21: Indicate the sampling height in Malmo.

**Answer:**

We have added the sampling height of 20 m.

Reviewers Comment: Page 6, line 9: The value selected for $NO_2$ differs from the one reported by Beelen et al. (2014) in the study which also provided the $PM_{10}$ RR used here. This should be discussed in section 4.4.

**Answer:**

The choice of RR in Faustini et al. (2014) instead of using Beelen et al. (2014) have been motivated in section 4.4.

Reviewers Comment: Figures 3-4: There are some extremely low mean monthly values for $PM_{10}$ in Gothenburg and $NO_x$ -$NO_2$ in Malmo. Provide an explanation, is it meteorology or else?

**Answer:**

For $NO_x$ and $NO_2$, we found some erroneous data that explained the low values. Therefore, Figure 4 has been replaced. The median trend is unaltered, while the upper and lower confidence interval only decrease by one hundredth $\mu g\ m^{-3}$. For $PM_{10}$, the low values are due to the way that the deseasonalisation is modelled. When we disregard the deseasonalisation, these extreme low values are not present. But we have decided to keep the deseasonalisation of the trend, in order to be consistent with all the other calculations.

Reviewers Comment: Table 1: The Table repeats the information of Figures 2-4. It should be removed altogether.

**Answer:**

OK, Table removed.

Reviewers Comment: Section 4.1.: The section needs an overhaul. The discussion should be performed by pollutant at the first level and then city specific mentions should be made where important differences occur. More clarity is needed in the argumentation. The potential impact of regional emission reductions from non-transport sources should be incorporated in the discussion.

**Answer:**

Section 4.1 has been revised substantially. The discussion is based primarily on the pollutants and secondarily on the cities. The potential impact of regional emission reductions from non-transport sources has been incorporated

Reviewers Comment: Page 16, line 12: Correct the phrasing here. Also, it is not clear how the dieselization has led to the reduction of $NO_x$ emissions. Krecl et al. (2017) report that $NO_x$ have remained constant during the process. Please elaborate. An indication of the change in the vehicle parc composition should be given in number.

**Answer:**

We have changed this and refer to statistics on diesel shares from BilSweden (2018).

Reviewers Comment: Page 16, lines 14-15: How does the location affect the trend? Indicating the distances from major roads could be informative, although the reported sampling heights are probably too large to represent direct road traffic emissions.

Answer:

We agree, and we have removed this.

Reviewers Comment: Page 16, lines 20-23: Given the site types and the sampling heights, it is somewhat doubtful that the primary $NO_2$ variability could be captured.

Answer:

Yes, we agree, and this is also clearly stated in the text in Section 4.1.1.

Reviewers Comment: Page 17, line 21: The whole discussion regarding the increasing ozone trend is obviously founded on the assumption that photochemical processes for ozone production in Sweden should be of minor importance. Otherwise the reduction of precursor emissions would generally lead to the long-term reduction of ozone as well, as it has been observed in various studies. Please, better clarify the dominant mechanism explaining the $O_3$ presence in the urban setting.

**Answer:**

The concentrations of ozone are lower at central urban background sites in the cities compared to outside the cities at rural background sites. This means that the net effect of the photochemistry involving ozone in the cities is that ozone is consumed, mainly due to the titration involving $NO_x$.

Reviewers Comment Page 17, lines 29-31: Given that $PM_{10}$ has been associated with vehicular emissions, shouldn't a similar to $NO_x$ weekday-weekend pattern be observed? Justify this difference.

**Answer:**

The main local source of $PM_{10}$ is road wear and road dust suspension (clearly seen in Stockfelt et al., 2017 and Segersson et al., 2017). Since the emissions of road dust strongly depend on the wetness of the roads, as shown by Johansson et al. (2007) and Denby et al. (2013), the diurnal cycles will not follow the same pattern as vehicle exhaust from traffic.

Reviewers Comment Page 18, lines 7-10: The hypothesis for the whole period cannot be supported by just one year of data and moreover these regional background data should be better described. Also, it is not clear why there aren't long term regional background NOx data available, when at Page 17, line 32 the availability of such data for $NO_2$ is stated.

**Answer:**

This part is rewritten with a new Section 4.1.1. Local and non-local contributions are explained more detailed, underpinned with several new references. The regional background stations are described in Section 4.1.2.

Reviewers Comment Page 20, line 24-34: The study which provided the RR for $PM_{10}$ includes similar results on mortality associations for $PM_{coarse}$. This could be discussed.

**Answer:**

Since $PM_{10}$ to a large extent consists of mechanically generated coarse particles in the Swedish cities, the similar results on mortality associated with $PM_{coarse}$ in Beelen et al. (2014) provide increased support for the RR that has been used. We have added a few sentences about this in Section 4.4, where RR associated with $PM_{10}$ exposure is discussed.

**Technical corrections:**

Page2, line 7: Check phrasing ("ending").

The phrasing has been changed.

Page 2, line 11: Check phrasing, you could replace with ". . .of the apparent major health impact of exposure to air pollutants...".

It is changed to "associated with"

Page 2, line 14: Delete "amount of".

Deleted

Page, 2, line 16: Equals sign not in subscript.

Changed

Page 4, line 13: . . .represents the urban background.

"the" urban is inserted.

Page 4, line 13-18: Remove the coordinate information.

Removed.

Page 5, line 21: Delete "decreasing".

Deleted.

Page 6, line 22: Replace "increase" with "change"

We don't want to change increase to "change", because it becomes illogical in relation to the following sentence, where the word decrease has been used.

Page 7, lines 25-30: This information is already provided in the Figure caption. Remove accordingly from the text. Figure 5: Ensure that Malmo is spelled consistently throughout the manuscript.

Lines 25-30 are removed, and the spelling Malmo is used consistently, including Figure 5.

Page 14, lines 8-9: Check phrasing. It should be "If the change in $O_3$ was only associated with local NO titration. . .".

Increase is changed to "change".

Page 16, line 10: Decrease and diminish, pick one.

The whole section 4.1 has been changed.

Page 19, line 14: Can be compared or can't compare? Please rephrase the sentence.

Can is changed to "cannot".

Page 19, line 16: Delete "extra".

Deleted

Page 20, line 12: "exposure occurs simultaneously"? What is meant here? Probably it refers to peak exposures. The same in section 4.5.

It means that environmental exposures to $NO_2$ and NO usually occur simultaneously, since the urban air contains both of these pollutants in varying proportions. Determining the effect of each pollutant can therefore be difficult. In Section 4.5 we clarify that double calculations regarding change in life expectancy occur if the effect of $NO_x$ and $NO_2$ is summarized, but this is not the case for the others pollutants, where the effects are assumed to be independent of each other.

Page 20, line 15: "Especially difficult is it. . .". Correct wording.

This part has been removed

Page 20, line 23: "Observed materials"?

It is changed from "less observed materials" to "less amount of data"

Page 21, line 28: You mean the population distribution within the cities?

Yes, "data" is changed to "distribution".

Page 22, line 29: $NO_2$ is formed by the reaction of nitric oxide with $O_3$.

Reformulated.

Figure A1-A4: Correct decimal separators.

The commas have been changed to point characters.

[revised manuscript text omitted]

---

## Author Response (AR2)

**Would like to thank the reviewer for the job that has been done with the review in the case of constructive improvement proposals. Below follow the comments and our answers highlighted in yellow, and the manuscript with track changes.**

**General comments**

The authors have addressed all raised issues and have substantially expanded the discussion in their manuscript, which now I consider much improved. A few final points to correct before proceeding with publication.

**Specific comments**

Figures 2-4: Maybe you could specify that the monthly average values are after deseasonalization.

It is added in the captions to Figures 2-4 that the monthly averages are based on deseasonalization.

Page 17, line 19: Check parentheses in the Johansson citation.

It is changed to Johansson et al. (2008).

Page 17, line 23: …trend diminishes…

Changed to diminishes.

Section 4.1: A reasonable answer has been provided in the response, regarding different weekday-weekend variability between NOx and PM10. See if you can incorporate this along with the references in the manuscript.

The following sentences have been incorporated in section 4.1.3: The pronounced weekday-weekend pattern associated with $NO_x$ and $NO_2$ is not shown for $PM_{10}$. Despite that $PM_{10}$ is mainly related to traffic, other factors also affect this pattern. Since the emissions of road dust highly depend on the wetness of the roads, as shown by Johansson et al. (2007), the diurnal cycles will not follow the same pattern as vehicle exhaust from traffic.

Page 18, line 10: Replace "photochemistry" with "ozone cycle".

Photochemistry is replaced with ozone cycle.

Page 18, line 20: Replace "perfect" with "pronounced".

Perfect is replaced with pronounced.

Page 20, line 25: The argument regarding passive vs. active sampling is fair. However, when the authors claim that the ESCAPE field campaigns were of limited extent or even inconsistent, the context has to be explained. I presume that they refer to the infrequent sampling. However, these data have been adjusted to annual means following a specific methodology. They should bear in mind that the same schedule was followed for PM sampling as well, that provided the PM10 RR that they have used in this study. I suggest to change the wording here.

Regarding the ESCAPE Study, we consider passive NO2 sampling less valid than PM monitoring with Harvard impactors. Since we don't have data on PM absorbance, we remove the sentence in section 4.4 "In addition, Beelen et al. (2014) did not find any significant association with PM2.5 absorption, possibly indicating problems to reflect exposure to vehicle exhaust". Further down in section 4.4, where the choice of RR for PM10 is explained, the sentence is rewritten as: For PM10, we use the RR value of 1.04 (95 % CI 1.00–1.09) per 10 μg m-3 increase from Beelen et al. (2014), where the meta-analysis in ESCAPE is based on measurement campaigns with Harvard impactors.

Page 21, line 16: Replace "accuracy" with "suitability".

Accuracy is replaced with suitability.

Page 23, line 10: …sectoral…

It is changed to sectoral.

Page 24, line 16: "…where NO has the capability of scavenging free oxygen atoms wherein NO2 is formed..". Rewrite this as: "..where NO oxidizes to NO2.." or omit the sentence. The titration reaction is between NO and O3 to form NO2.

It is rewritten.

Figure A1-A4: Please also correct separators for the axis values.

The separators are corrected.

[revised manuscript text omitted]